# A reduced state of being: The role of culture in illness perceptions of young adults diagnosed with depressive disorders in Singapore

Wen Lin Teh[1]*, Ellaisha Samari[1], Laxman Cetty[1], Roystonn Kumarasan[1], Fiona Devi[1], Shazana Shahwan[1], Nisha Chandwani[2], Mythily Subramaniam[1]

1 Research Division, Institute of Mental Health, Singapore, Singapore, 2 Department of Mood and Anxiety, Institute of Mental Health, Singapore, Singapore

* Wen_Lin_Teh@imh.com.sg

## Abstract

Illness perceptions form a key part of common-sense models which are used widely to explain variations in patient behaviours in healthcare. Despite the pervasiveness of depressive disorders worldwide and in young adults, illness perceptions of depressive disorders have not yet been well understood. Moreover, while a high proportion of cases of depressive disorders reside in South-east Asia, few have explored illness perceptions that are culturally relevant to this region. To address these limitations, this study aimed to understand illness perceptions of young adults diagnosed with depressive disorders. Face-to-face semi-structured interviews were conducted among Chinese, Malay, and Indian young adults aged 20 to 35 years old, who were seeking treatment at a psychiatric hospital. Data reached saturation after 33 interviews (10 to 12 interviews per ethnic group) and five themes emerged from the thematic analysis: 1) A reduced state of being experienced at a point of goal disengagement, 2) the accumulation of chronic stressors in a system that demands success and discourages the pursuit of personally meaningful goals, 3) a wide range of symptoms that are uncontrollable and disabling, 4) poor decision making resulting in wasted opportunities, with some positive takeaways, and 5) accepting the chronicity of depression. Young adults typically experienced depression as a reduced state of being and it was thought of cognitively as an entity that may be a part of or separate from the self. Over and beyond these aspects of cognitive representations was the emergence of themes depicting conflicts and dilemmas between the self and the social environment that threatened self-identity and autonomy. Addressing these conflicts in therapy would therefore be of utmost relevance for young adults recovering from depressive disorders in the local setting.

## Introduction

The term *illness* evokes not only the experience of symptoms and suffering, but also the judgments, interpretations, coping, and help-seeking behaviours that go with it [1]. The construction of illness, as explained by the theory of social constructionism, is shaped, and influenced

**Data Availability Statement:** We have indicated that the data for this study is available upon request. The restriction is imposed by our institutional and ethics committee. Data can only

be shared after a proposal is approved by the ethics committee. This has also been conveyed to our participants during the consent process. The data request can be sent to The Institutional Research Review Committee, Institute of Mental Health, Singapore; Email address: imhresearch@imh.com.sg.

**Funding:** This research is supported by the Singapore Ministry of Health's National Medical Research Council under the Centre Grant Programme (Grant No.: NMRC/CG/M002/2017_IMH). The funders had no role in study design, data collection and analysis, decision to publish, or preparation of the manuscript.

**Competing interests:** The authors have declared that no competing interests exist.

by sociocultural forces, providing social meaning to the experience of a disease [2]. In the context of illness and health, illness narratives serve as the most instinctive way of communicating the social construction of diseases [1, 3].

Illness narratives form a key part of explanatory models or common-sense models, which are used widely to explain variations in patient behaviours in healthcare. Examples of such models include the health belief model [4], the theory of planned behaviour [5], and the theory of self-efficacy [6]. One popular model is the Common-Sense Model of Self-Regulation (CSM) by Leventhal, which posits that individuals are active problem-solvers who try to understand their illness so as to cope with it [7–9]. Individuals do so by formulating their own set of cognitive representations of illness to self-regulate and manage health threats associated with their illness [7]. The central aspect of CSM is the use of 'lay' beliefs to predict health-promoting or coping behaviours.

According to the model, illness representations of any illness comprise of at least five dimensions: 1) Identity: symptoms that are perceived to be part of the illness, 2) Time-line: acute or chronic in duration, 3) Consequences: the extent of the impact on individuals' lives, 4) Causes: whether the illness is due to internal or external causes, and 5) Cure or control: the extent to which the illness is curable or incurable. The model also consists of two independent processes (cognitive and emotional) that deal with the representation of fear or danger of illness threats which trigger coping mechanisms and outcome appraisals thereafter providing feedback to the model [7]. The CSM is one of the most widely used, due to its good psychometric properties and applicability in predicting self-care behaviours and recovery in physical illnesses [10, 11].

Increasingly, CSM is being applied to mental illness research as a way to understand and predict patient behaviours. Unlike other health belief models, CSM comprises of emotional illness representations, which are directly relevant to mental illnesses, making it a useful guide [12]. In their review article, Lobban and colleagues detailed numerous studies that were consistent with the CSM and had described how the model is able to explain and fit the mental illness experience [12].

However, several authors have highlighted limitations over the applicability of CSM in mental illness as opposed to physical illnesses [12–14]. While CSM is a useful guide, it has been criticized to be overly simplistic in explaining illness representations of mental illnesses [12, 15]. Unlike physical illnesses, the cause-and-effect associations in mental illnesses are far less clear as there are greater variations in the causal beliefs and experiences of mental illnesses. Individuals often struggle to accept mental illness [13, 16, 17]. Coming to terms with the illness is neither straightforward nor coherent [13, 18], such that individuals may express different illness beliefs simultaneously even if those beliefs contradict each other [14]. Moreover, unlike the case of physical illnesses where there is a clear distinction between the self and illness, the boundary between mental illness and the self is far less clear and often intersects. As a result, individuals with mental illnesses do not always see their symptoms in a negative light. Higbed and Fox's qualitative investigation into thirteen patients with anorexia nervosa, reported that individuals who felt the condition was a meaningful part of their identity were reluctant to do away with the illness [19]. Pedley and colleagues' qualitative investigation into sixteen people with obsessive-compulsive disorder found that participants perceived their symptoms positively as a sort of peculiarity in their personality, rather than viewing them negatively as symptoms of an illness [20]. Thus, unlike physical illnesses, mental illness symptoms are not always undesired and this has crucial ethical implications in the context of treating mental disorders where the patients' definitions of recovery are often overlooked.

Yet consolidating an explanatory model for mental illness remains a challenge as patient narratives differ considerably by the type of illness. Perceptions of more common forms of

mental illnesses, such as depression or anxiety, are more varied as compared to less common forms, such as schizophrenia [14], with the former conditions being more dependent on what is considered to be culturally accepted expressions of "distress" [14]. For instance, Asian patients with depression tend to describe illness complaints as physical rather than psychological, such as reporting to feel 'tired' rather than 'sad', since the psychological labelling of symptoms is often stigmatized in non-western cultures [21]. In another example, mental illness is more likely to be attributed to a 'weakness of character' in Japanese individuals as compared to Australian individuals, who are more likely to attribute it to physiological reasons [22]. These examples depict the role of culture, its influence and interaction with personal beliefs in the social construction and experience of illness [23].

Till date, the majority of qualitative work have been conducted amongst severe but less common forms of mental illnesses, such as psychosis or schizophrenia [12, 14], while qualitative investigations into more common forms of mental illnesses, such as depressive disorders are few and far between. Within the context of depressive disorders, the majority of existing studies have been conducted among individuals with ongoing primary physical conditions who also report depressive symptoms, patients who meet criteria for depression based on self-report and cut-off scores, or among healthcare professionals [24–27]. Yet, only a few have been conducted amongst individuals who are formally diagnosed with depressive disorders and who are seeking psychiatric care [28, 29].

The World Health Organization (WHO) estimates that approximately 4.4% of individuals suffer from depression worldwide [30]. According to the same report, the majority of cases reside in South-east Asia (27% of cases), followed by the Western Pacific region (21% of cases), Eastern Mediterranean region (16% of cases), the Americas (15% of cases), European (12% of cases), and African regions (9% of cases) [30], yet few studies have explored illness perceptions that are culturally relevant to this region. Furthermore, understanding the culture surrounding young adulthood ensures that mental health services stay relevant to their needs; needs which have grown more complex today than ever before [31]. Young adulthood is arguably the period where an individual is most productive in life and from an economic standpoint, mental illness can pose a significant threat to productivity and disease burden. This is crucial for Singapore as human capital is considered to be an important asset [32]. Major Depressive Disorder (MDD) is the most pervasive disorder among young adults aged 18 to 34 years in Singapore [33] and has a significant treatment gap [34]. Thus, understanding how illness is experienced among young adults would be important for interventions at the initial stages of disease onset. To the best of our knowledge, there has been no qualitative inquiry into illness perceptions of depression that is specific to a non-western psychiatric population in Singapore and while it is typically assumed that non-western communities have more spiritual/traditional narratives of mental illness [21, 22, 35], a highly globalized multi-ethnic society such as Singapore may present illness narratives that are culturally distinct; it is not uncommon for Singaporeans to simultaneously utilize both allopathic and traditional forms of treatment in a multifarious medical system [36–38].

The present study aims to explore illness representations in individuals with depressive disorders in Singapore. The two main research questions are: 1) What are the illness perceptions of young persons with depression in Singapore? and 2) How does the role of culture influence illness perceptions in young people diagnosed with depression locally?

While the nature of our inquiry is exploratory, we would expect illness perceptions to be highly associated with the self. Additionally, these dimensions are also expected to be more complex, reflecting a combination of both allopathic and culturally derived concepts of illness representation.

## Methods

### Procedure

Convenient and purposive sampling were utilized for recruitment. Recruitment was conducted in two ways: individuals were primarily referred to the research team by their clinicians during their outpatient visit or inpatient stay or were approached at the outpatient clinics by a member of the research team with information flyers that described the study. The inclusion criteria: Singapore citizens or permanent residents, aged 18 to 35 years, with a Diagnostic and Statistical Manual, Fourth Edition (DSM-IV) diagnosis of depressive disorder, receiving outpatient treatment at the Institute of Mental Health (IMH), and were able to speak, read, and write in English. The exclusion criteria: patients with a diagnosis of substance-induced depressive disorders, depressive disorder due to a general medical condition, post-natal depression, bipolar disorder, and depression with psychotic symptoms.

All consent taking procedures (both verbal and written consent) were conducted by members of the research team and not by their referring clinicians to prevent coercion. Participants who verbally agreed to be interviewed were re-contacted within a few days but no longer than a week later via phone contact. Patients were then asked a second time over the phone if they were willing to be interviewed. This ensured that participants had sufficient time and space to reconsider their participation. After the second verbal consent was given, the researcher scheduled an interview appointment with the participant. Inpatients who agreed to the study were interviewed after they were discharged. Written consent was obtained from all participants before the start of the interviews.

As the communication styles of individuals in Asian cultures differ substantially from Western cultures, one-to-one interviews were preferred over focus group discussions [39]. To minimize the possibility that participants may describe their illness beliefs in medical terms, the interviews were carried out at the research interview rooms which were situated away from the clinical setting in IMH. Only one interview was conducted outside of the hospital at the convenience of the participant. To minimize the potential lack of cultural sensitivity that may present itself during the interviews, interviewees were matched with interviewers according to ethnicity as much as possible. An interview schedule (S1 File) that covered questions on the depression experience and culture was used. Interviews were conducted primarily in English; however, as participants were bilingual, they were encouraged to converse in their mother tongue languages if they felt more comfortable doing so. Interviews were audio-recorded and transcribed verbatim in both English and mother tongue languages (Chinese/Malay/Tamil) which were translated in the same transcript. Interviews lasted approximately 1 hour and 3 minutes on average, and the duration of the interviews ranged from approximately 33 minutes to 2 hours.

This study was approved by the local institutional ethics and review boards: the Domain Specific Review Board (DSRB) of the National Health Group (NHG) and the Institutional Research Review Committee (IRRC) in IMH.

### Participants

A total of 52 individuals were approached from February 2018 to January 2019. Fourteen of those approached had refused to participate or were uncontactable when re-contacted. Most individuals did not cite any reason for refusing, while some others mentioned that they were uninterested, busy, or uncomfortable with the study's interview format. Individuals signed written informed consent forms before the start of the interviews. Diagnoses were initially based on self-report or by a referring clinician (NS, who knew the inclusion and exclusion

criteria for this study) and were only verified after the completion of the study with their electronic medical records (upon getting written consent to do so) or their doctors. Four participants were withdrawn as they did not fulfil the eligibility criteria. Thus, a total of 34 individuals (19 females and 15 males) with depressive disorders completed the interviews. Subsequently, one transcript was excluded from the analysis as it differed substantially from the rest of the transcripts (participant's illness beliefs were primarily associated with another medical condition), leaving a final count of 33 transcripts. Participants were purposively sampled to ensure adequate representation by ethnic groups. Data reached saturation after 12 Chinese, 11 Malay, and 9 Indian participants were interviewed. Although one participant was Sri Lankan, she had been living in Singapore most of her life. Her narrative met with great familiarity and was analysed with the rest of the transcripts. The average age was 26 years old and ranged from 20 to 35 years old at the point of the interview. Participants were on average living with depressive disorders for approximately 3.7 years, ranging from 4 months to 16 years. In terms of highest educational qualifications attained, 12 participants had pre-university or secondary school qualifications and 21 participants had a diploma, university degree, or vocational qualifications. All of the participant characteristics are summarized in Table 1.

## Analysis

We chose to analyse the transcripts with thematic analysis described by Clarke and Braun [40, 41] because of the advantages it offers to researchers working in the healthcare setting. Thematic analysis allows researchers in healthcare to overcome methodological constraints, such as a lack of time, resources, and expertise that are required for many other qualitative approaches. The authors (WLT, ES, LC, RK, FD, SS) are all quantitative researchers by formal training (i.e., degrees in psychology/sociology at least), have all been trained in qualitative research, and are experienced in the use of thematic analysis.

The initial coding process involved conducting open coding of each sentence in each transcript [42, 43]. The first author utilized deductive (themes of CSM were determined a priori) and inductive (from ground up) coding simultaneously throughout the coding process using the framework described by Feredey and Muir-Cochrane in 2006 [44]. Inductive coding involved recognizing, identifying, and clustering of important information that represented aspects of a phenomenon. On the other hand, deductive coding involved using pre-existing codes as a guide and template for the clustering of information. Transcripts were analysed using the Nvivo software.

All transcripts were analysed thematically by the first author (WLT) to determine if 1) data reached theoretical saturation for each ethnic group and if 2) there were new emergent codes or themes. The initial process of coding started at the midpoint of the study as interviews were being conducted. Transcripts were initially analysed by ethnicity, but as the preliminary themes did not differ substantially between ethnic groups, transcripts were eventually analysed as a whole. A codebook that comprised of the codes and its' definitions was reviewed and analysed iteratively through multiple discussions between the team members (WLT, ES, LC, RK, FD, SS) until a high inter-rater agreement of 0.73 (Kappa) was achieved. Co-author MS provided advice and guidance over the study design, procedures, and analyses of the study. Interviews ceased when no new information, codes, or themes emerged from the interviews that could change the codebook, in other words, data had reached theoretical saturation [43]. The first author also wrote memos detailing her immediate thoughts of possible links to theoretical concepts during the entire thematic process, which were later discussed alongside the review of the codebook with team members.

**Table 1. Self-reported participant characteristics and sociodemographic information (n = 33).**

| S/N | Age | Sex | Ethnicity | Years[1] | Highest education | Marital Status | Religion | Work Status |
|---|---|---|---|---|---|---|---|---|
| C01 | 34 | F | Chinese | 8 | Polytechnic Diploma | Single | Buddhism | Unemployed |
| C02 | 20 | F | Chinese | <1 | A' level/Completed Pre-U or Junior College | Single | Christianity | Student |
| C04 | 30 | M | Chinese | 2 | Polytechnic Diploma | Single | Buddhism | Full-time |
| C05 | 34 | F | Chinese | 7 | University degree | Single | Christianity | Part-time |
| C06 | 26 | M | Chinese | 8 | O/N level/ Completed Secondary education | Single | Christianity | Part-time |
| C07 | 24 | F | Chinese | <1 | Polytechnic Diploma | Single | Taoism | Part-time |
| C08 | 31 | F | Chinese | 3.5 | University degree | Single | Christianity | Unemployed |
| C09 | 35 | F | Chinese | <1 | Polytechnic Diploma | Married | Buddhism | Full-time |
| C10 | 22 | F | Chinese | 4 | A' level/Completed Pre-U or Junior College | Single | Christianity | Student |
| C11 | 22 | M | Chinese | 2 | A' level/Completed Pre-U or Junior College | Single | Buddhism | Student |
| C12 | 22 | F | Chinese | 2^ | A' level/Completed Pre-U or Junior College | Single | Buddhism | Student |
| C13 | 20 | M | Chinese | 3 | A' level/Completed Pre-U or Junior College | Single | Free-thinker | Other |
| L01 | 34 | F | Indian | 16 | University degree | Single | Hinduism | Part-time |
| L02 | 27 | M | Indian | 7^ | Polytechnic Diploma | Single | Buddhism | Part-time |
| L03 | 27 | F | Indian | 4 | University degree | Single | Hinduism | Unemployed |
| L05 | 21 | M | Indian | 1 | O/N level/ Completed Secondary education | Single | Free-thinker | Other |
| L06 | 26 | M | Indian | 2 | Polytechnic Diploma | Single | Christianity | Unemployed |
| L08 | 22 | F | Indian | <1 | Polytechnic Diploma | Single | Free-thinker | Full-time |
| L09 | 26 | F | Indian | 8 | University degree | Single | Islam | Full-time |
| L10 | 25 | M | Indian | 2 | Polytechnic Diploma | Single | Others | Student |
| L11 | 23 | M | Indian | 7 | O/N level/ Completed Secondary education | Single | Christianity | Unemployed |
| L12 | 22 | F | Sri Lankan | 6 | O/N level/ Completed Secondary education | Single | Free-thinker | Student |
| M01 | 26 | M | Malay | 3 | Polytechnic Diploma | Single | Islam | Full-time |
| M02 | 23 | M | Malay | 1 | Polytechnic Diploma | Single | Islam | Full-time |
| M03 | 27 | M | Malay | <1 | Polytechnic Diploma | Single | Islam | Full-time |
| M04 | 21 | M | Malay | 1 | Vocational Institute/ ITE Nitec Cert | Single | Islam | Student |
| M05 | 28 | M | Malay | 2 | University degree | Single | Islam | Other |
| M06 | 22 | F | Malay | 4 | Polytechnic Diploma | Single | Islam | Part-time |
| M07 | 25 | F | Malay | <1 | O/N level/ Completed Secondary education | Single | Free-thinker | Full-time |
| M09 | 35 | F | Malay | 2 | Polytechnic Diploma | Single | Others | Full-time |
| M10 | 29 | F | Malay | 2 | O/N level/ Completed Secondary education | Single | Islam | Full-time |
| M11 | 31 | F | Malay | 3 | O/N level/ Completed Secondary education | Married | Islam | Full-time |
| M12 | 26 | F | Malay | 1 | Polytechnic Diploma | Single | Islam | Full-time |

Note: Years[1] denote years diagnosed with depressive disorder; F is female; M is male; ^means an approximate number of years diagnosed with depressive disorder self-reported by the participant.

# Results

The themes that emerged were generally in accordance with the CSM as determined a priori. However, additional themes emerged from inductive coding which were separate from the concept of CSM. In summary, five major themes emerged: 1) A reduced state of being experienced at a point of goal disengagement, 2) The accumulation of chronic stressors in a system that demands success and discourages the pursuit of personally meaningful goals, 3) A wide range of symptoms that are uncontrollable and disabling, 4) Poor decision making resulting in wasted opportunities, with some positive takeaways, and 5) Accepting the chronicity of depression.

## 1. A reduced state of being experienced at a point of goal disengagement

**1.1 At a point of disengagement of existing (unattainable) goals.** Depression was likened to "a stop button in my life." It was experienced as a point of change or disengagement from the lack of achievement of existing goals and expectations.

> "It has put like a stop on life for me. I'm, I've stopped going to school, I've stopped just doing anything I liked, and just kept to myself at home. It just, yeah it was like a stop button in my life." O12/22/F/SL

> "...So I would say prior to when I was depressed I was kind of interested in joining the public service so I was really kind of structuring my life, my interests, a lot of things lah[1] to fit within that idealized career or mine... It really changed my whole direction in life lah[1]. For now, I really, I really don't want to join the public service... I've just completely given up that dream." C11/22/M/C

> [1]**lah** *and loh are Singlish (Singapore English) colloquial words that function as discourse particles that set the tone of conversations (usually to stress a point) but do not alter the meaning of sentences.*

It manifested cognitively as a lack of direction (i.e., feeling lost and stuck) and motivation (i.e., having no point in life), and affectively as frustration and discontentment.

> "Or...oh ya like feeling trapped or like they do the job, same thing every day like the life not going anywhere kind of thing." M02/23/M/M

> "...you fail to see the meaning of a lot of things like...maybe there's no reason to live, there's no reason to work, there's no point in doing all these because...nothing will come out of it..." I06/26/M/I

**1.2. An entity that is separate or a part of the self.** Depression was described to be associated with the self in two ways. First, depression was seen as a separate entity that endangered the sense of self. Some participants were constantly battling depression in their minds, preventing it from encroaching into their sense of selves. Experiencing a depressive episode was thus seen as being powerless over the hold that depression had over them which they depicted as an indomitable force.

> "I'm constantly fighting my demons so I'm very small and my demons are this big. And they're loud and they're strong so on bad days I was just tell my friends lah like my demons are here today I don't wanna talk to anyone. So that's how I relate my bad thoughts and and and... they too strong for me sometimes that I can't even hear my own voice" M07/25/F

> "No actually it's not really about being sad... So *[the thing in]* the mind is taking over your body... So you can't, you can't actually fight whatever that you're feeling. So you just let the feeling come and stay for a while..." M11/31/F/

Depression had a lasting impact on self-identity. Over time, it displaced the self.

> "...I don't even know myself anymore... 'cause all my friends all say ... "why you become like that, errr I want the old [C07] back' but every single time they say that I don't even recall how I was last time. I don't even feel like human... As time pass it just get... it just eats you up..." C07/24/F/C

Second, depression was seen as part of the self by some participants, primarily as an emotion that everyone experiences in varying degrees.

"Because depression, in itself, is an emotion that everyone goes through. Yeah, but they may be able to get out again before it becomes like a mental health condition per say. "I09/26/F/I

As these participants perceived symptoms of depression to be universally present, the meaning of "having depression" had more to do with taking on the medical label rather than the presence of it.

". . .it's just the more depressed me, side of me talking. . .I think I actually had it already depression. As in there wasn't a name to it when I had it. . . I'm like oh yeah, like it's an official medical term to it. Then ok, I have it." C12/22/F/C

Reactions to the medical label were mixed. Being clinically diagnosed was reassuring as the feelings that they had been dealing with for a long period of time were legitimized, yet at the same time, there was a sense of guilt as having the medical label provided a justification for a lack of 'control' over these symptoms.

"Sometimes it can be comforting knowing that there is a, I would say medical term to it. . . Sometimes when I feel like I told you, feel lazy about it. Ok just let it be. And let the depression sink in. So, sometimes when you have a medical term to it where you know what your diagnosed with, you your own self might take it, take advantage of the term that you have and you let yourself spiral down also." C12/22/F/C

**1.3. A reduced state of being.**    Regardless of whether depression was seen as separate from or as a part of the self, participants generally believed that a depressive episode was a reduced state of being that eventuated from a depletion of cognitive resources that were used to deal with external (i.e., environmental or social) afflictions which was usually narrated to be chronic in nature. In other words, depression is experienced when the mind was no longer "strong enough" to withstand periods of accumulated stress.

"I was still like high functioning kind of thing but perhaps they just went on uh untreated for too long and the various disorders and symptoms or syndromes just kept accumulating and getting more and more serious you know so perhaps it's like a, like maybe my brain just can't take it any more lah that kind of thing. . ." C08/31/F/C

"I think it makes sense that it should be a. . . it would be (an) accumulation of factors because we're all build in a certain amount of resilient you know and then when it just gets too much then you become depressed I suppose. . ." I01/34/F/I

## 2. The accumulation of chronic stressors in a system that demands success and discourages the pursuit of personally meaningful goals

**2.1. Causes are multi-faceted and often chronic in nature.**    While participants acknowledged that there are multiple biological, environmental, and social causes, which could interact and lead to depression, participants reiterated that a large part of it was due to an accumulation of chronic environmental or social stressors. Family was frequently seen as a main source of stress, conflict, instability, and abuse which panned out over a long period of time.

"... I think when we were 13 the whole lot of us including the first wife and his 5 sons were in like a house together and I was just like trying so hard to be happy, like playing games with them but I know this isn't right you know. Like they're constantly being so impulsive and like running and making the wrong decisions and impacting everyone under them." M07/25/F

As a result of the chronic lack of warmth and closeness with the family, some individuals became heavily and emotionally invested in relationships with significant others. Unfortunately, when these relationships dissolved, it became overwhelming, triggering episodes of depression.

"... So when we separated, part ways, um I guess my emotions...I needed emotional support elsewhere. Because she was my support... not my mum, not my dad, not my sister...just her. So I felt myself going down and down and down and down until to the point I couldn't cope, I couldn't eat. You know because I was so reliant on her." I10/25/M/I

Additionally, work and school were cited as common sources of stress. Individuals were expected to perform or conform to a strict and highly pressured work culture or were overwhelmed by the demands of school. The inability to leave the chronic stressor (i.e., "They don't have a choice") and the lack of timely resolution was perceived to lead to an accumulation of stress past a threshold and into depression.

"Six o'clock... is the end of the day, nobody can leave at 6. There's once I leave at 6:15, they call me back just to do another work because I leave too early." C07/24/F/C

"...it's being constantly uhm, being put into a circumstance or a position where you're not, one thing is that you're not happy. For a very, very long time and suddenly that becomes the new normal?" M05/28/M/M

**2.2. High societal expectations of success.** The "chase of achievements" or being an "overachiever" at school or work is a societal expectation and a norm that is instilled in the family since young. It is the "system that demands that I do well" (C11/22/M). This mentality was seen as damaging and was believed to be a main contributor of depression for some. Much of the content in negative thinking (i.e., feeling useless, shame) appeared to be associated with the lack of accomplishment.

"...The chase for achievements that you know that is very part of how we function as a country. Like most of the yah you know like from young it's tuition and whatever and like all sorts of enrichment classes and all that yah so like yah so like but actually that kind of mind-mentality yah . . . for someone with depression especially someone who has been you know always pushing and perhaps it's the pushing thing that just uh, that could be one of the causes..." C08/31/F/C

"... I didn't want to go to school, I didn't want to face exams. Then, I felt that if I didn't do the exams then, because at that time they say that if you do very well in your exams, get a good job and everything, then I felt that if I didn't go and face exams, I mean I won't be able to, I won't be able to clear the, get the A levels ah. Then I feel that I'm very useless, then I feel that maybe I should just die or something... I felt a sense of shame in myself. I cannot push myself to go and take the exams and I think, I think a lot of negative thoughts came back" C06/26/M/C

"... Like people's expectations and then when we cannot meet it, we feel like we failed, that constant feeling of failure over the years could... It just seems like something traumatic I guess, less intense but it builds over time, or like it's some sort of traumatic experience, bullying abuse, that kind of thing." I05/21/F/I

**2.3. Unable to pursue personal goals that are incongruent with familial and societal expectations.** According to some participants, depression was experienced when they felt they had failed to reach their goals. Living up to high familial/societal expectations was difficult, yet switching to goals that were personally more meaningful was not seen as an option for many. Not being able to define and pursue personal goals, especially goals that were incongruent with the expectations of the family or society, were articulated as a contributor to depression. The lack of agency in influencing the direction of their future that is personally meaningful left them feeling demotivated and trapped at the same time.

"So yeah it's every day studying something you have no interest in, and then knowing you're never going to use it? Or if you do it's going to be in a job that you don't like. So it sort of, just made me feel very down, miserable about it, and sort of angry at everybody that I was wasting my time when I could be learning something I felt passionate about." I05/21/M/I

"I think it's a sense of, don't have any motivation to do anything in life. Don't have a goal. Just do everything because... I don't see meaning in doing anything, I don't have a goal." C06/26/M/C

"...all my life you know throughout my secondary school I just never did what I wanted to la. Like the CCA I wanted to, I wanted to do a third language I also wasn't allowed to. So my life was kinda decided for me you know kinda thing ... I would lose interest in it in fact you know and I just become disinterested. I still have that a bit now ya but at that point it was more pronounced la. So you know I just didn't had the courage to stand up to them (family) I guess." I01/34/F/I

## 3. A wide range of symptoms that are uncontrollable and disabling

Participants mentioned somatic, psychological, cognitive, and emotional symptoms as indicators of depression that were disabling in nature. Participants cited somatic symptoms most often. These symptoms included sleeping issues such as insomnia, hypersomnia, low appetite, low energy, fatigue, or migraines. Behavioural symptoms such as self-harm and attempted suicide were also mentioned but less frequently.

"If not one day you don't want to be awake. You just... you just don't want to do anything. You just want to sleep throughout your entire life." I01/34/F/I

"I never experienced the sadness symptoms at all, for me it's just I just feel tired loh[1] exhausted loh or empty or I feel nothing." C08/31/F/C

"Low energy is constant; I feel this all the time, low self-esteem" C10/22/F/C

In addition to behavioural and somatic symptoms, participants commonly cited cognitive symptoms. The two types of cognitive symptoms that were typically present in the narratives: 1) Cognitive symptoms that were overly active, uncontrollable, and intrusive (i.e. rumination or catastrophic thinking), and 2) cognitive aspects which were absent or reduced (i.e. lacking motivation, lacking concentration, helplessness or hopelessness).

"I don't know why my memory, my brain very active. I try to like, it's very tiring cause it's a, I told you it's a loop. So, legit, the scene will just play out very fast. . ." C12/22/F/C

". . .the ability to concentrate. . .um, the self-confidence that you have in yourself and this sense of satisfaction which I feel that depression takes away. . . Even after completing a very large project, I don't actually feel that sense of satisfaction which makes it very hard for me to keep going and do like the next thing and the next thing because I never feel fulfilled, or never feel satisfied or this sense of achievement." I09/26/F/I

Participants described psychological and emotional symptoms as well, such as feeling sad, crying, anxiety and hallucinations, but were mentioned less frequently.

## 4. Poor decision making resulting in wasted opportunities, with some positive takeaways

For the majority of participants, depression had a negative impact on their general and social functioning, and school or work performances. In addition, some participants recounted making poor decisions on key instances which adversely affected their future outlook.

**4.1. Consequences from poor decision making and wasted opportunities.** Several participants felt that because of depression, they held themselves back from key opportunities that would otherwise improve future prospects if they had seized them. Interestingly, not being able to take on opportunities was indicative of being separate from normal.

"I will say it makes me more like. . .it makes me feel like I am separated from others and I don't really have the opportunities that other people have. Opportunities more like I am holding myself back, that kind of like my school has this erm. . .not event more like a group called the [PROGRAMME] where it's basically like all the people with high GPA go into and . . .like because of depression, I can't really put myself in there" M04/21/M/M

**4.2. Positive consequences.** Contrastingly, a few participants believed that there were positive consequences. One participant believed depression gave her a sense of empathy and advocacy for others with mental illnesses.

". . . I could kind of erm empathize with them to a high degree because I could look at the person and say look you know I understand what you going through and really advocate for them, push for them without any judgment you know and I think that definitely helped a lot so my depression would certainly have helped in that. . ." I01/34/F/I

## 5. Accepting the chronic nature of depression

Participants had initially believed that depression was an acute condition that could be cured within a short amount of time but most have come to accept that depression is a chronic condition, and the route to recovery is long and arduous.

". . .actually last time, I thought that depression will last for like a few years, but now I've actually accept that it will last uh for a lifetime. Yah. Because uh not matter how we fight it, if the person around me keeps on giving you negativity, you will still stick to your depression. If the things that triggers in the first place just won't go off, then the sickness will always stay. . ." M11/31/F

"Like all along I kind of uh was viewing this whole uhm journey of you know journeying through my various health conditions and all I kind of just viewed as it like a phase and I just can't wait for it to be over. But when I saw that (an Instagram post) and like, it kind of spoke to me in that yah perhaps uhm it's actually even ok if this so called dark period is not just a period, but just. . . there. I would just be in this darkness uh. . . for as long as it is." C08/31/F/C

Depression was likened to cancer. The majority of participants did not believe that depression could be eradicated completely or that they could be fully cured of its symptoms (i.e. "Not a hundred percent curable"). Depression could be managed or treated but the chance of relapse was thought of as highly likely due to unforeseen stressors that may crop up in the future.

"As far as I'm concerned it's something that I'll have for the rest of my life and I just, at this point I'm just managing it at the moment" M05/28/M/M

"I don't know if there is a cure for it right. I mean, I don't know. There's only the treatment I guess and it's like cancer, can come back anytime" M10/29/F

From the narratives, depression was thought of cognitively as an entity that may be a part of or separate from the self. Regardless of cognitive representations, depression was primarily *experienced* as a reduced state of being that occurred as a result of a depletion of resources to cope with unresolved and often chronic stressors. Such chronic stressors typically comprised having to deal with prolonged periods of strained relations with the family, or demands from school or work. Having depression meant making poor decisions on key instances which participants perceived to have had profound long-term consequences. Further, depression was seen as a chronic condition with no means of a full recovery. Over and beyond these aspects of cognitive representations was the emergence of themes surrounding goal disengagement and re-engagement which was unique to the present narratives.

## Discussion

This study aimed to explore illness perceptions in young adults with depressive disorders in Singapore. The themes that emerged from this study were to an extent similar to past literature supporting the claim that the CSM has its limits in modelling the way individuals perceive mental illnesses due to added dimensions associated with the self [12, 14]. Along with these results, there were also emergent themes that were unique to depressive disorders which will be discussed at greater length in the present section.

### Depression, self-identity, and goal disengagement

Based on the themes that emerged from inductive open coding, depression was highly associated with the self and self-concepts. Our findings support the notion that there might be an added dimension of illness perception regarding the extent to which the condition is seen as a part of or separate from the self [20], and could reflect differing levels of conflict or acceptance during identity reconstruction [45, 46].

Unlike physical illnesses, which are almost always appraised as unwanted external afflictions to the body, individuals may or may not see depressive disorders in the same regard. When individuals were asked what depression meant to them, they generally described it as a psychological entity that was either separate from or a part of the self, which is consistent with past reports that investigated illness perceptions in individuals with severe types of mental

illnesses [13, 14, 17, 19, 20]. Due to the realization that depression is a chronic condition with no means of full recovery, young adults saw themselves grappling with identity changes manifesting within the self [46]. Regardless, episodes of depression were often described as a reduced state of being from their normal selves. This experience of depression is similar to that found in a qualitative study by Kinderman et al. among individuals with psychosis, where episodes were considered to be "periods of socially recognized states of altered psychological functioning" [13]. However, there is a difference. According to participants, a reduced state of being was experienced as a result of a depletion of cognitive resources required to cope with the accumulation of unresolved and often chronic stressors, which is vastly different from being simply in an altered state of irrationality described by individuals with psychosis [13]. Finally, similar to past reports, depression is not always appraised as negative [19, 20]. Having depression gave new meaning and perspectives (e.g., advocacy, empathy) into the lives of a few participants.

From our findings, conflicts in self-identity are central to the experiences of depression. Constructivist theorists have posited that individuals experience implicative dilemmas, defined as a type of cognitive dissonance that arises whenever a desired change in a self-concept clashes with a system's worldview, on which such a change is undesired [47]. Conflicts within self-concepts in relation to goal reconstruction seem to reflect a struggle to satisfy society's desired ideals alongside personal aspirations.

Even though Singapore is often lauded as a melting pot of eastern and western cultures, the family is fundamentally collectivistic in nature. Mainstream ideals of success that are propagated from Chinese influences (the majority ethnic group) are inculcated in the family and in school. Capitalistic values of competitiveness and meritocracy, fuelled by the need to survive as a nation since the industrial era, are entrenched in the cultural fabric of Singapore [48]. From a young age, individuals are taught at home and in school to be successful and to be a useful member of society; life's happiness is contingent on a person's level of contribution to society which is measured by achievements and social standing. The majority of young adults recalled feeling that they were under tremendous pressure to fulfil unrealistic ideals [49] originating from the family [50]. Additionally, at least relevant for the current sample, this pressure was contributed by an achievement-focused mind-set that was heavily instilled from a young age, and consistently reminded in schools and workplaces which demand excellence and uncompromised productivity. Participants described this mind-set to be detrimental when goals became unattainable; the pressure to succeed at such goals in a system that demands success, could have contributed to the rise of more severe forms of depression [51, 52]. The perceived inability to re-engage new and personally meaningful goals could have further exacerbated and prolonged symptoms of depression past a clinical threshold [53, 54]. Moreover, it was evident that the difficulties surrounding goal re-engagement were partly due to the fear of negative social ramifications (e.g., shame, conflict, social exclusion), as the expectations of the family are almost non-negotiable in typical Asian families [55, 56]. Contemplating to reconstruct personally meaningful goals that may go against society's ideal expectations of success seem to parallel a clash of worldviews, which in turn, can create high levels of conflict within the self [47].

Coping with goal failure is one aspect that is rarely discussed among clinical populations [51, 52]. The link between goal disengagement/engagement and the depression experience found in this study would thus warrant future research into this phenomenon. Cognitive paradigms suggest that depression can be improved by addressing the negative cognitive triad (negative view of the self, the world, and the future) in therapy [57]. Consequently, dilemma-focused therapy is as beneficial as Cognitive Behavioural Therapy (CBT) in ameliorating symptoms of depression [58]. Our results support the notion that addressing implicative dilemmas

in addition to negative cognitive self-concepts is important and greatly relevant culturally for the recovery of depression in young adults in Singapore [59]. Given that the average onset of depression lies in the period of young adulthood [60], which is also a crucial period of psycho-social development that includes identity formation, self-development, and achievement goal orientations [61–65], the temporal overlap with such crucial developmental period further exacerbates the need to understand the role of culture and family in goal reconstruction in young adulthood. Discussing cultural expectations of success with the family in therapy would thus be beneficial for young adults with depression locally.

## Depression illness beliefs in a multi-ethnic non-western society

Culture was initially assumed in this study as static and pre-defined identities that tie closely to ethnicity (i.e. Chinese/Malay/Indian cultures). However, the similarities in narratives across ethnic groups reflect a different definition of culture, or macro-culture, defined as how individuals actively make sense of their lives and illness through common experiences regardless of ethnic background [66]. Illness perceptions of depressive disorders were generally similar across Chinese, Malay, and Indian ethnic groups, and to that of the West [17, 18, 67]. Perhaps due to the sample's young adult makeup that consisted of a highly educated majority (67% of participants had a diploma or university degrees), causal beliefs were neither overwhelmingly spiritual nor traditional but were highly similar to themes reported in the extant literature that also had a strong emphasis on socio-environmental and psychosocial causes [13, 18, 20, 68]. This was accompanied by a preponderance of the belief that a relapse was strongly dependent on unforeseen future socio-environmental triggers that could potentially provoke future episodes of depression.

Collectively, such beliefs are noteworthy as two inferences can be derived. First, even as young adult patients sought biomedical interventions, such as pharmacotherapy, they were perceived to work primarily as a temporary relief of symptoms rather than a panacea to rid the condition altogether [13]. Second, it is perceived that the root of the depression lies in the disharmony of the self within the system, and therefore, it cannot be fully addressed by engaging either one part in isolation. Recovery of depression should be approached by addressing both cultural and biopsychosocial factors in an integrated fashion.

An integrated approach should be emphasized as the country gains momentum in reshaping its healthcare system towards one that is patient centred. While young adults do benefit from clinical interventions, their narratives paint a broader picture of how limiting these interventions may be without considering the sociocultural context. Depression means to be in a reduced state of being, which is a frame of mind of living a less than meaningful life, characterised by a struggle to reconcile intersubjective misalignments [69] that threatens self-identity and autonomy. Recovery thus not only involves a return to premorbid health but also involves the process of regaining control over one's agency, goals, and aspirations [70]. Our findings further support the notion that depression is caused (in part) by the strained relationship between the self and the social environment [45]. Therefore. regaining control over goals would therefore require the recognition and support from the immediate social environment [69, 71], on which a young self-identity can continually be validated and fostered.

This study is not without limitations. First, some of the participants were undergoing treatment and/or therapy at the time of the interview and it is unclear how much of illness beliefs were a result of psychoeducation. Second, we recruited young adults who were willing and able to sit through an interview. It is possible that those who refused the study, especially those who had cited that they did not feel comfortable with the interview format, may have different illness experiences. Thus, illness beliefs found in this study may differ from those who are not undergoing formal treatment or who refused the study.

On the surface, young adults' illness representations of depression have struck a chord with past literature. Careful thematic explication revealed important sociocultural nuances that were unique to the current context. Young adults reportedly faced tremendous pressure to fulfil unrealistic societal expectations and had consequently struggled to construct personally meaningful goals. Depression is thus a frame of mind of living a less than meaningful life, which at a latent level, represented an underlying struggle of addressing implicative dilemmas that represent a clash of personal and cultural worldviews. These findings re-emphasized the role of culture in illness perceptions of depression and self-identity. Addressing societal and familial expectations in relation to goal failure would therefore have immense relevance for young adult individuals seeking psychotherapy in Singapore.

## Supporting information

**S1 File. Interview schedule.**
(DOCX)

## Author Contributions

**Conceptualization:** Wen Lin Teh, Fiona Devi, Shazana Shahwan, Nisha Chandwani, Mythily Subramaniam.

**Data curation:** Wen Lin Teh, Ellaisha Samari, Laxman Cetty, Roystonn Kumarasan, Fiona Devi, Shazana Shahwan, Nisha Chandwani.

**Formal analysis:** Wen Lin Teh, Ellaisha Samari, Laxman Cetty, Roystonn Kumarasan, Fiona Devi.

**Funding acquisition:** Wen Lin Teh, Mythily Subramaniam.

**Investigation:** Wen Lin Teh, Ellaisha Samari, Laxman Cetty, Roystonn Kumarasan, Shazana Shahwan, Nisha Chandwani.

**Methodology:** Wen Lin Teh, Laxman Cetty, Roystonn Kumarasan, Fiona Devi, Shazana Shahwan.

**Project administration:** Wen Lin Teh, Ellaisha Samari, Laxman Cetty, Nisha Chandwani.

**Resources:** Wen Lin Teh, Nisha Chandwani, Mythily Subramaniam.

**Software:** Wen Lin Teh, Ellaisha Samari.

**Supervision:** Shazana Shahwan, Mythily Subramaniam.

**Validation:** Wen Lin Teh, Mythily Subramaniam.

**Visualization:** Wen Lin Teh.

**Writing – original draft:** Wen Lin Teh.

**Writing – review & editing:** Wen Lin Teh, Ellaisha Samari, Laxman Cetty, Roystonn Kumarasan, Fiona Devi, Shazana Shahwan, Nisha Chandwani, Mythily Subramaniam.

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
