## [Decision Letter · Decision Letter 0]

12 Oct 2020

PONE-D-20-07514

A reduced state of being: illness perceptions in young adults diagnosed with depressive disorders

PLOS ONE

Dear Dr. Teh,

Thank you for submitting your manuscript to PLOS ONE. After careful consideration, we feel that it has merit but does not fully meet PLOS ONE’s publication criteria as it currently stands. Therefore, we invite you to submit a revised version of the manuscript that addresses the points raised during the review process.

We look forward to receiving your revised manuscript.

Kind regards,

Stephan Doering, M.D.

Academic Editor

PLOS ONE

2. Please provide further detail regarding the interviews that were conducted as part of this study.

In the Methods section please outline the topics covered in the interview.

Please state whether an interview guide was used. If so, please include a copy of the topic/interview guide used in the study, in both the original language and English, as Supporting Information, or include a citation if it has been published previously.

3. Please provide additional details regarding participant consent.

In the Methods section, please state why it was not possible to obtain written consent, how verbal consent was recorded and whether the ethics committee approved this consent procedure.

If your study included minors, state whether you obtained consent from parents or guardians.

5. Please amend your manuscript to include your abstract after the title page.

Reviewers' comments:

Reviewer's Responses to Questions

**Comments to the Author**

1. Is the manuscript technically sound, and do the data support the conclusions?

Reviewer #1: Partly

Reviewer #2: Yes

2. Has the statistical analysis been performed appropriately and rigorously? 

Reviewer #1: N/A

Reviewer #2: N/A

3. Have the authors made all data underlying the findings in their manuscript fully available?

Reviewer #1: No

Reviewer #2: Yes

4. Is the manuscript presented in an intelligible fashion and written in standard English?

Reviewer #1: Yes

Reviewer #2: Yes

5. Review Comments to the Author

Reviewer #1: This paper presents an original qualitative research on illness perceptions in young adults diagnosed with depressive disorders based on an Asian context (Singapour). Overall, the topic is highly relevant for psychology and the qualitative perspective s most appropriate to explore narratives to understand the lived experience of people with depressive disorders. The paper is well-structured and the English is correct.

Despite these qualities, the current version of the paper calls for substantial changes at epistemological, theoretical and methodological levels in order to be potentially ready for publication in PLOS ONE, especially given the solid and high quality of this journal. Here below my suggestions to improve the quality of the present version of the manuscript:

1) Introduction: The Leventhal model is presented in the introductory part as a core theoretical perspective mobilized by the authors, as they position themselves critically towards such model. However, the reader wonders why, given the broad body of qualitative research on depression, the introduction is narrowed down to research on this model? What about other qualitative approaches to depression, that also take into account narratives on illness (ex. phenomenology and IPA, narrative, discursive...).

I very much appreciated that you mention the role of culture as it shapes the experience of illness - This is an inherent part of the socio-constructionist perspective namely. But scientific references are missing to better argue on the crucial role of culture (understood as a set of norms and values - giving rise to social narratives that contribute to determine how we experience the world and ourselves). The distinction between West and East remains rather superficial and broad. What seems specific to the context you chose for your research and why is it relevant for the hypothesis that you raise at the end of this section?

Furthermore, while narratives seem the crucial focus on this paper, the reader cannot really grasp from the manuscript's content why the narrative focus is so important. As a qualitative researcher in psychology, I personally fully agree with you on the central role of narratives. But why are narratives so important? Different perspectives have been developed within psychology, namely that of structuring our sense of selves and our world (discursive, constructionist, narrative), but also that of making sense of our experience (for example, phenomenology). However, the choices underlying your theoretical approach seem absent /scarcely developed. Indeed, there are multiple approaches to narratives - and this theoretical foundation is currently missing from your paper. The problematisation at the end of the introduction falls short: why the hypothesis on the sense of self? Can you develop what you mean by this and argue with additional references to make your research/theoretical posture more explicit?

2) Methodology: To me, this section requires important changes. Under the 'procedure' paragraph, the population is already mentioned but then you come back to participants in the next paragraph entitled 'participants' - The reader can be a bit lost in this order. I would suggest to better justify the qualitative approach used in the 'procedure' paragraph to better explain the whole methodological procedure in a general way.

line 119: what do you mean by 'subconsciously'? not appropriate from my perspective - better explain your decision on the context of the interviews

In the population paragraph, the average age needs to be better justified - (especially if 'young adults' is part of your title - why young adults rather than older adults? Also, there is no mention of the gender except for the number of male and female - but according to me, the 'gender' dimension needs to be at least considered and justified to increase methodological rigor - as in depression this seems to play an important role (ex. constructions of masculinity/feminity)

Also, on the interview description - there is no information on the interview guide nor how it was elaborated. This is an important aspect to better understand how the analysis were conducted afterwards, in the analytical phase.

Regarding the 'analysis' section - practically no scientific references are used to justify your analysis technique. Yet, this methodological stage is particularly important. I would suggest reading Virginia Braun and Victoria Clarke who have extensively written on thematic content analysis.

Line 144: Saturation? Better explain and use scientific reference (This notion is contested in qualitative research depending on the authors).

Line 158: open and ground coding - what does this mean? Reference please

3) Results: The themes are very general - A 'good' theme definition is that which captures in its name the full condensed 'story' or 'message' that the given set of data portrays.

Moreover, please restructure with numbers or letters in hierarchical order the different themes and subthemes as it is difficult for the reader to follow. I regret that there is a lot of 'verbatim' that illustrate the names of your subthemes but little explanation or interpretation on what these themes and subthemes mean in relation to your hypothesis and research aim. At times, the text is written as if narratives 'reflected' a 'truth' or a 'reality' - Please keep in mind what status do narratives have in your theoretical perspective to better guide you on how to write about them.

I sometimes disagreed with the authors' explanation on certain quotes - to me, Line 200 for instance, would call for a separation between mind and body rather than depression as a different entity, from what I understand from the participants' verbatim

Often, authors refer to results as 'they' (participants') as if all participants had positioned themselves in an homogeneous way regarding their experience. Were there any dissident narratives for example?

Line 286: while this seems a key theme (or subtheme?) it is very little developed/explained.

Line 315: spelling of 'yeah'

Line 352: This theme follows a hierarchical numeration while the rest of the themes do not.

In the end of the result section, a synthesis of results would be very useful.

Discussion: Coming back to the CSM model - why and how it was confirmed? Why do authors overlook the similarities of their results with the model (if it's an important focus of the paper)?

Line 428 'are under pressure' - this is a strong affirmation that is different from 'perceive themselves as being under strong pressure' for example. This study is qualitative so it takes the subjective constructions into account - not revealing an 'objective' reality.

Line 474: regarding the limitations, please consider that the aim of qualitative research is not generalization but on the contrary, considering variablity, contextualization, narratives, singularity, experience - therefore it is not a limitation from my view.

Finally, how can your conclusions be extended beyond the Leventhal model and contribute to psychology more broadly? This could be a stronger case if your theoretical position/framework were better made explicit in the introduction for example.

Reviewer #2: Thank you for asking me to review this manuscript.

This qualitative inquiry aimed to explore the illness perceptions of Chinese, Malay, and Indian young adults diagnosed with depressive disorders by using one-to one face-to-face semi-structured interviews. They concluded that depression was typically experienced as a reduced state of being, and was thought of cognitively as an entity that may be a part of or separate from the self. Five themes were identified as: 1) meanings, 2) causes, 3) symptoms, 4) consequences, and 5) chronicity of depression.

Such an effort could be valuable for cultural diversity and future implication. The following issues need to be considered:

1. In abstract, the research purpose is expected to be added.

2. In abstract, Conclusion-“Depression was typically experienced as a reduced state of being, and was thought of cognitively as an entity that may be a part of or separate from the self. .. the results emphasized the importance of examining self concepts in therapy and recovery.…” Such a conclusion might be relevant to young adults’ developmental task at their developmental stage? It might not be applicable for other groups at different ages?

3. In Introduction, please add the rationale why selecting the young adults as a target? Relevant research significance would be helpful, eg; What’s the global prevalence of major depression in young-aged papulation?

4. The positive aspects of the illness perception were mentioned from several literatures (Line 48-57). Please compare your results with previous studies in Discussion.

5. In Method, Line121-More clear information is needed, for example: interviewees’ qualification and training? Were interview process standardized or using the identical interview guidelines? What contents of interview guidelines included? Or how did you achieve the same focus of the interview contents?

6. How could the researcher prevent yourselves to be influenced or guided by these prior understanding? For example, the existing knowledge on the Common-sense Model of Self-Regulation (CSM) by Leventhal. Was the interview guideline or analytical coding process followed by the CSM theory?

7. Line144-150, please add the Standardized Deviation following the mean to indicate the variation and range of the sample characteristics. Most are single, please report the %.

8. Line148, 6 participants live with depressive disorders lower than 1 year, with a mean of 3.5 years, and the least was 4 months. How will it influence your results? It might be some limitation.

9. About the analysis, in Line 145-146- How was the transcript of one individual from Sri Lankan analyzed with the rest of the transcripts as a whole?

10. In Results, it’s not easy to distinguish the “themes” and “sub-themes”. Please clearly separate the “5-themes” and “sub-themes” to increase the readability.

11. Among all themes identified as cognitive representations, I’m wondering was there any emotional perspective or anything related to their coping or management with the mood symptoms while looking at their illness ration than cognitive aspects?

12. Line 356, “3.4.1…”?

13. Table—what’s the meaning of the 「^」 following the number, eg: 2^ or 7^?

6. PLOS authors have the option to publish the peer review history of their article (what does this mean?). If published, this will include your full peer review and any attached files.

Reviewer #1: **Yes: **Maria del Rio Carral

Reviewer #2: No

---

## [Author Response · Author response to Decision Letter 0]

25 Nov 2020

18 November 2020

RE: Responses to reviewers

Dear Reviewers,

We thank you for taking the time to review our manuscript. We have learnt a lot from the peer review process and have incorporated your comments in our next draft. Please find our point-by-point responses to the comments in the boxed text area below:

Reviewer #1: This paper presents an original qualitative research on illness perceptions in young adults diagnosed with depressive disorders based on an Asian context (Singapour). Overall, the topic is highly relevant for psychology and the qualitative perspective s most appropriate to explore narratives to understand the lived experience of people with depressive disorders. The paper is well-structured and the English is correct.

Despite these qualities, the current version of the paper calls for substantial changes at epistemological, theoretical and methodological levels in order to be potentially ready for publication in PLOS ONE, especially given the solid and high quality of this journal. Here below my suggestions to improve the quality of the present version of the manuscript:

1) Introduction: The Leventhal model is presented in the introductory part as a core theoretical perspective mobilized by the authors, as they position themselves critically towards such model. However, the reader wonders why, given the broad body of qualitative research on depression, the introduction is narrowed down to research on this model? What about other qualitative approaches to depression, that also take into account narratives on illness (ex. phenomenology and IPA, narrative, discursive...).

Answer: Thank you for this. We agree with the reviewer that more could be elaborated on this point. There are important reasons for using Leventhal’s model. Firstly, the Leventhal model is one of the most widely used model due to its good psychometric properties, to predict self-care behaviours and recovery in physical illnesses. Secondly, unlike other health belief models, this model comprises of emotional illness representations, which are directly relevant to mental illnesses. We have elaborated on this point in the subsequent draft.

We agree that there are many other qualitative approaches that are highly relevant to the study of narratives. For instance, IPA is very useful in providing rich accounts of the depression experience. However, as it had been previously discussed by Braun and Clarke (2020) and Chamberlain (2012), there is no one ideal or superior qualitative method. The choice of method may also be dependent on pragmatic factors as well- we are unfortunately limited in our access to resources and are facing time constraints. Finally, authors (and coders of this study) are experienced and trained in thematic analysis only. We wish to be able to use other qualitative approaches in the near future because of its relevance in the work that we do. We agree with the reviewer that this was insufficiently justified in the manuscript and have made substantial edits to it.

I very much appreciated that you mention the role of culture as it shapes the experience of illness - This is an inherent part of the socio-constructionist perspective namely. But scientific references are missing to better argue on the crucial role of culture (understood as a set of norms and values - giving rise to social narratives that contribute to determine how we experience the world and ourselves). The distinction between West and East remains rather superficial and broad. What seems specific to the context you chose for your research and why is it relevant for the hypothesis that you raise at the end of this section?

Answer: Thank you for this. As the reviewer had succinctly pointed out, the illness experience of an individual may be influenced by societal norms and values. We agree fully with the reviewer that this portion is underdeveloped in the present draft, and have since made changes in the subsequent draft.

Furthermore, while narratives seem the crucial focus on this paper, the reader cannot really grasp from the manuscript's content why the narrative focus is so important. As a qualitative researcher in psychology, I personally fully agree with you on the central role of narratives. But why are narratives so important? Different perspectives have been developed within psychology, namely that of structuring our sense of selves and our world (discursive, constructionist, narrative), but also that of making sense of our experience (for example, phenomenology). However, the choices underlying your theoretical approach seem absent /scarcely developed. Indeed, there are multiple approaches to narratives - and this theoretical foundation is currently missing from your paper. The problematisation at the end of the introduction falls short: why the hypothesis on the sense of self? Can you develop what you mean by this and argue with additional references to make your research/theoretical posture more explicit?

Answer: As quantitative researchers by training, we are largely unfamiliar with the theoretical concepts that are widely used in sociology. However, social constructivism, as we recognize, is highly relevant to our study. Thus, we understand and agree fully with the reviewer that the introduction can be improved further by laying theoretical grounds for the study and we hope the subsequent draft reads better. We thank the reviewer for these suggestions.

2) Methodology: To me, this section requires important changes. Under the 'procedure' paragraph, the population is already mentioned but then you come back to participants in the next paragraph entitled 'participants' - The reader can be a bit lost in this order. I would suggest to better justify the qualitative approach used in the 'procedure' paragraph to better explain the whole methodological procedure in a general way.

line 119: what do you mean by 'subconsciously'? not appropriate from my perspective - better explain your decision on the context of the interviews

Answer: Thank you for this. The main reason for carrying out the interviews away from the clinical setting was to prevent contextual priming at a subconscious level that may influence the way participants describe their illness experience clinically during the interviews. We have rephrased this explanation and restructured the paragraphs. 

In the population paragraph, the average age needs to be better justified - (especially if 'young adults' is part of your title - why young adults rather than older adults? Also, there is no mention of the gender except for the number of male and female - but according to me, the 'gender' dimension needs to be at least considered and justified to increase methodological rigor - as in depression this seems to play an important role (ex. constructions of masculinity/feminity)

Answer: Major Depressive Disorder is the most prevalent mental illness among young adults aged 18 to 34 years old (Subramaniam et al., 2020), thus, these narratives describe how depression is typically experienced at the onset for this age group. Understanding how illness is experienced among young adults would be important for treatment and recovery that is also relevant for this age group. 

Before the start of the study, researchers conducted a literature review of illness perception of severe mental illnesses. As reported in the introduction, the primary themes that emerged were related to the self, with no prominent mentions of gender. During the coding and analyses phase, the first author did suspect possible cultural influences on the role of gender. For instance, the theme “High societal expectations of success”, initially emerged from the narratives of male participants. This was detailed in her memo, indicating that instances like this could demonstrate cultural expectations of the need to succeed being more salient in men. However, during the course of coding, this theme was prominent in female participants as well. While we agree with the reviewer that gender does play an important role, it did not emerge during the literature review and analyses.

Also, on the interview description - there is no information on the interview guide nor how it was elaborated. This is an important aspect to better understand how the analysis were conducted afterwards, in the analytical phase.

Answer: Thank you for this, the interview guide has been attached as supporting information in the subsequent submission.

Regarding the 'analysis' section - practically no scientific references are used to justify your analysis technique. Yet, this methodological stage is particularly important. I would suggest reading Virginia Braun and Victoria Clarke who have extensively written on thematic content analysis. Line 144: Saturation? Better explain and use scientific reference (This notion is contested in qualitative research depending on the authors). Line 158: open and ground coding - what does this mean? Reference please

Answer: Thank you for this. We recognize that this is important but was mistakenly left out. We have uploaded the interview schedule in its final form. We have also improved on the analysis section to explain our sampling and analysis technique, including scientific references to support our justifications.

3) Results: The themes are very general - A 'good' theme definition is that which captures in its name the full condensed 'story' or 'message' that the given set of data portrays.

Moreover, please restructure with numbers or letters in hierarchical order the different themes and subthemes as it is difficult for the reader to follow. I regret that there is a lot of 'verbatim' that illustrate the names of your subthemes but little explanation or interpretation on what these themes and subthemes mean in relation to your hypothesis and research aim. At times, the text is written as if narratives 'reflected' a 'truth' or a 'reality' - Please keep in mind what status do narratives have in your theoretical perspective to better guide you on how to write about them.

Answer: Thank you for the suggestions. We have made additional changes to the name of the themes/subthemes to better represent the themes that emerged from the narratives.

I sometimes disagreed with the authors' explanation on certain quotes - to me, Line 200 for instance, would call for a separation between mind and body rather than depression as a different entity, from what I understand from the participants' verbatim

The verbatim quote may not have clearly captured the context. What we had understood from the participant during the interview was that “…the thing is in your mind. So [the thing in] the mind is taking over your body…” M11/31/F/M

Often, authors refer to results as 'they' (participants') as if all participants had positioned themselves in an homogeneous way regarding their experience. Were there any dissident narratives for example?

Answer: Thank you for this. Only the prominent themes were reported in the manuscript. The prominence of themes was not necessarily dependent on the majority although it was primarily so, and we would have stated if the themes or codes emerged only from a few. For instance, the narrative that depression is separate from or a part of the self, came from two different groups of participants, which we had merged to form a subtheme. We did encounter dissident narratives, for instance, narratives like, “depression comes from the brain so I think to actually rewire something takes a long time M12/26/F”, or “depression is something that’s very, very repetitive C12/22/F”, but we did not report as it was decided among coders that they were not prominent codes.

Line 286: while this seems a key theme (or subtheme?) it is very little developed/explained.

Line 315: spelling of 'yeah'

Line 352: This theme follows a hierarchical numeration while the rest of the themes do not.

In the end of the result section, a synthesis of results would be very useful.

Answer: Thanks for highlighting as these were mistakes. We have edited/removed them in the subsequent draft. The format of the manuscript has changed, and we hope it will read better.

Discussion: Coming back to the CSM model - why and how it was confirmed? Why do authors overlook the similarities of their results with the model (if it's an important focus of the paper)?

Thank you for this. The main objective of the study is to understand illness perceptions of depressive disorders, and it was not to disprove or confirm the CSM model. We had however, made few concluding statements as our study provided support for the conclusion that the CSM may be overly simplistic, although it is a useful guide for mental illnesses. 

Line 428 'are under pressure' - this is a strong affirmation that is different from 'perceive themselves as being under strong pressure' for example. This study is qualitative so it takes the subjective constructions into account - not revealing an 'objective' reality.

Line 474: regarding the limitations, please consider that the aim of qualitative research is not generalization but on the contrary, considering variablity, contextualization, narratives, singularity, experience - therefore it is not a limitation from my view.

Answer: We agree with the reviewer and had removed this statement from limitations. We thank the reviewer for this important point.

Finally, how can your conclusions be extended beyond the Leventhal model and contribute to psychology more broadly? This could be a stronger case if your theoretical position/framework were better made explicit in the introduction for example.

Answer: The themes that emerged in this study were highly associated with the self, emphasizing the importance of self-concepts, such as self-agency, goals, sense of meaning and belonging, in the study of illness perceptions of severe mental illnesses. Additionally, our results evidently show that personal recovery requires the consideration of cultural beliefs that tie closely to illness beliefs among those surviving with depressive disorders in Singapore. 

Thank you for taking the time to review this article. We hope the subsequent draft reads better.

 

Reviewer #2: Thank you for asking me to review this manuscript.

This qualitative inquiry aimed to explore the illness perceptions of Chinese, Malay, and Indian young adults diagnosed with depressive disorders by using one-to one face-to-face semi-structured interviews. They concluded that depression was typically experienced as a reduced state of being, and was thought of cognitively as an entity that may be a part of or separate from the self. Five themes were identified as: 1) meanings, 2) causes, 3) symptoms, 4) consequences, and 5) chronicity of depression.

Such an effort could be valuable for cultural diversity and future implication. The following issues need to be considered:

1. In abstract, the research purpose is expected to be added.

Answer: It has been added. We thank the reviewer for spotting this missing point.

2. In abstract, Conclusion-“Depression was typically experienced as a reduced state of being, and was thought of cognitively as an entity that may be a part of or separate from the self. .. the results emphasized the importance of examining self-concepts in therapy and recovery.…” Such a conclusion might be relevant to young adults’ developmental task at their developmental stage? It might not be applicable for other groups at different ages?

Answer: Thank you for this. We agree with the reviewer that individuals go through different developmental stages throughout their lives and the conclusion is only relevant to young adults, which is the premise of this study. Major Depressive Disorder is the most prevalent mental illness among young adults aged 18 to 34 years old (Subramaniam et al., 2020), thus, these narratives are descriptive of how depression is typically experienced at the onset for this age group. This aspect of the abstract has been changed to elaborate this point.

3. In Introduction, please add the rationale why selecting the young adults as a target? Relevant research significance would be helpful, eg; What’s the global prevalence of major depression in young-aged papulation?

Answer: Thank you for this. Local data reports that the onset of depressive disorders lie within the age range of 18-34 years, thereby making it a relevant age range to target. By understanding how depression is experienced among young adults, it may help to inform treatment and therapy that is relevant for this age group. We agree with the reviewer that this is underdeveloped in the original draft, and have made substantial improvements. We hope the subsequent draft reads better.

4. The positive aspects of the illness perception were mentioned from several literatures (Line 48-57). Please compare your results with previous studies in Discussion.

Answer: Thank you for this. Similarly seen in past reports on anorexia nervosa and obsessive compulsive disorder, depression was not always appraised as negative (Higbed & Fox, 2010; Pedley, Bee, Wearden, & Berry, 2019)). Having depression gave new meaning and perspectives (i.e. advocacy, empathy) into the lives of some participants. We agree with the reviewer that this portion should be included in the discussion.

5. In Method, Line121-More clear information is needed, for example: interviewees’ qualification and training? Were interview process standardized or using the identical interview guidelines? What contents of interview guidelines included? Or how did you achieve the same focus of the interview contents?

Answer: Thank you for this. The interviewers and coders were trained in thematic analysis conducted by the National University of Singapore (NUS). The interview process was standardized using an interview schedule, which we had left out during the first submission; we have included the interview guide as supporting information in the subsequent draft.

6. How could the researcher prevent yourselves to be influenced or guided by these prior understanding? For example, the existing knowledge on the Common-sense Model of Self-Regulation (CSM) by Leventhal. Was the interview guideline or analytical coding process followed by the CSM theory?

Answer: Thank you for this. Before the start of qualitative inquiry, the first author and coders conducted preliminary literature review searches to attain a level of theoretical sensitivity (Mills, Bonner, & Francis, 2006). The interview guide thus follows the CSM theory.

7. Line144-150, please add the Standardized Deviation following the mean to indicate the variation and range of the sample characteristics. Most are single, please report the %.

Answer: Thank you for this. We have reported the SD and percentages in the following draft.

8. Line148, 6 participants live with depressive disorders lower than 1 year, with a mean of 3.5 years, and the least was 4 months. How will it influence your results? It might be some limitation.

Answer: Thank you for the question. We understand the reviewer’s concerns: an individual who was officially diagnosed just 4 months ago would have a different experience than another who had been officially diagnosed 16 years ago. In this study, outpatients were invited to participate regardless of the length of years diagnosed. We did not put limits on this, as deciphering the exact start points of mental illness (unlike physical illness which are more clear) is difficult unless individuals were followed through a prospective longitudinal study. Furthermore, it is not uncommon for individuals to experience a duration of untreated depression for a good number of years without their awareness, before an official diagnosis is given. Thus, this information may be obscure as well. Finally, we did not find any deviations in narratives of those who lived with depressive disorders for less than 1 year than the rest. 

9. About the analysis, in Line 145-146- How was the transcript of one individual from Sri Lankan analyzed with the rest of the transcripts as a whole?

Answer: That participant, though Sri Lankan, had mentioned in the interview (and recorded in the transcript) that she had “lived in Singapore most of (her) life”. Her narrative was not substantially different from the rest, and hence, her transcript was analysed as a whole since she did not belong to any of the major ethnic groups. That is, during the meetings between coders, the themes that were coded in her transcript was analysed and reviewed together with the rest of the transcripts while ignoring her Sri Lankan ethnicity.

10. In Results, it’s not easy to distinguish the “themes” and “sub-themes”. Please clearly separate the “5-themes” and “sub-themes” to increase the readability.

11. Among all themes identified as cognitive representations, I’m wondering was there any emotional perspective or anything related to their coping or management with the mood symptoms while looking at their illness ration than cognitive aspects?

Answer: Thank you for this, questions on coping methods and management were indeed asked during the interviews (please see the interview schedule). However, we had decided to discuss this portion in a separate manuscript.

12. Line 356, “3.4.1…”?

Answer: This was an error. Thank you for spotting this.

13. Table—what’s the meaning of the 「^」 following the number, eg: 2^ or 7^?

The symbol “^” denotes self-reported approximate number of years diagnosed with depressive disorder. These participants could not remember the exact year they were diagnosed, hence, the figures reported are a rough estimation.

Answer: Thank you very much for taking the time to review this article.

--

We hope the subsequent draft reads better and we are open for further discussions and feedback. We hope to hear from you soon and thank you very much again for the time to review our manuscript.

Your sincerely,

Wen Lin

Institute of Mental Health Singapore

 

References cited:

Higbed, L., & Fox, J. R. (2010). Illness perceptions in anorexia nervosa: A qualitative investigation. British Journal of Clinical Psychology, 49(3), 307-325. 

Mills, J., Bonner, A., & Francis, K. (2006). The development of constructivist grounded theory. International journal of qualitative methods, 5(1), 25-35. 

Pedley, R., Bee, P., Wearden, A., & Berry, K. (2019). Illness perceptions in people with obsessive-compulsive disorder; A qualitative study. PloS One, 14(3), e0213495. 

Subramaniam, M., Abdin, E., Vaingankar, J., Shafie, S., Chua, B., Sambasivam, R., . . . Chua, H. (2020). Tracking the mental health of a nation: prevalence and correlates of mental disorders in the second Singapore mental health study. Epidemiology Psychiatric Sciences, 1-10.

---

## [Decision Letter · Decision Letter 1]

5 Feb 2021

PONE-D-20-07514R1

A reduced state of being: illness perceptions in young adults diagnosed with depressive disorders

PLOS ONE

Dear Dr.Teh,

Thank you for submitting your manuscript to PLOS ONE. After careful consideration, we feel that it has merit but does not fully meet PLOS ONE’s publication criteria as it currently stands. Therefore, we invite you to submit a revised version of the manuscript that addresses the points raised during the review process.

We look forward to receiving your revised manuscript.

Kind regards,

Stephan Doering, M.D.

Academic Editor

PLOS ONE

Reviewers' comments:

Reviewer's Responses to Questions

**Comments to the Author**

1. If the authors have adequately addressed your comments raised in a previous round of review and you feel that this manuscript is now acceptable for publication, you may indicate that here to bypass the “Comments to the Author” section, enter your conflict of interest statement in the “Confidential to Editor” section, and submit your "Accept" recommendation.

Reviewer #1: (No Response)

Reviewer #2: (No Response)

2. Is the manuscript technically sound, and do the data support the conclusions?

Reviewer #1: Yes

Reviewer #2: Yes

3. Has the statistical analysis been performed appropriately and rigorously? 

Reviewer #1: N/A

Reviewer #2: N/A

4. Have the authors made all data underlying the findings in their manuscript fully available?

Reviewer #1: No

Reviewer #2: Yes

5. Is the manuscript presented in an intelligible fashion and written in standard English?

Reviewer #1: Yes

Reviewer #2: Yes

6. Review Comments to the Author

Reviewer #1: Thank you for responding to my comments. While the new version is substantially better, in my view there are still important aspects that are problematic, from an epistemological point of view. In addition, I still have some concerns on the compatibility of theory (based on a model stemming from a rather cognitive tradition) and methodology (based on understanding of contextualized experiences). Last, the discussion remains too narrow (scientific and practical implications are scarcely discussed).

General comment on epistemological concerns

The use of qualitative approaches in psychology does not imply the ‘mere’ implementation of interviews + conducting a thematic content analysis, but rather requires an in-depth reflection on how personal experiences are shaped by the social and the cultural, as well as more psychological and embodied aspects. In order to analyse individuals’ experiences, qualitative researches need to de-construct certain presuppositions (e.g. when you state ‘Since illness beliefs are personal in nature’ – this is a very strong assumption that many qualitative psychologists would argue against – as many approaches show how individual experiences and subjectivity – including ‘beliefs’ are socially constructed and therefore contextual – thus the opposite of ‘personal in nature’)

Another example of ‘lack of coherence’ between incompatible paradigms is the following statement:

“From the perspectives of personality psychology and evolutionary biology, depressive symptoms are adaptive forms of coping that facilitates the detachment of existing unattainable goals to conserve resources which may otherwise further deplete such resources if individuals persisted with such goals (28-30).”

This purpose is not coherent with qualitative approaches as it ‘essentialises’ goal setting as if this skill was completely detached from social contexts, social discourses, etc.

General comment on the articulation of theory and qualitative methodology

As I have tried to explain, qualitative research tradition in psychology stresses the social embeddedness of psychological phenomena. Also, it requires to overcome the risk of methodolatry (see Chamberlain’s paper on this). There shall be more reasons than ‘simply’ practical or resource-related, to justify the use of a method rather than another one. Moreover, the difference that you state between interviews and focus groups is not necessarily true – many authors would argue that this statement

“Since illness beliefs are personal in nature one-to-one interviews are preferred over

focus group discussions as they provide the space to discuss personal attitudes without being pressured to give socially desirable answers” is not the difference between these two methods at all. As authors admit they come from more quantitative traditions, I would urge them to read qualitative research handbooks on these methods of data collection (and foremost, data analysis methods as they scarcely quote any authors in their analysis technique description)

General comment on scientific/broader implications

The discussion remains descriptive and too narrow – the reader may ask her/himself ‘so what?’ – what are the implications of studying individual experiences of depression in Singapour? As I had suggested in my previous review, the paper would improve its potential if it could broaden its scope from a theoretical perspective (What do your findings mean for theory on depression (Western-based) and furthermore, what about more practical implications for people who suffer from depression, for healthcare, for psychologists – namely in non Western contexts like Singapour.

Reviewer #2: PONE-D-20-07514 A reduced state of being: illness perceptions in young adults diagnosed with depressive disorders

Thank you for asking me to secondly review this manuscript. Most of the comments has been well responded, but the following issues still need to be considered to make their effort valuable in cultural diversity and future implication.

1. I’m curious could it be possible that the illness perceptions and experience of your participants have not limited to the CSM framework? The author replied that their interview guide follows the CSM theory, that will easily produce an accordant result with the CSM as their prior framework. How could the researcher prevent yourselves to be mainly influenced by the prior understanding on the existing knowledge on the CSM?

2. The stigma-related issues in non-western society have been mentioned in Discussion; but it has not been reflected from any of patients’ quotations?

3. Please clarify some unclear wordings in Discussion, such as:

…that was heavily instilled “from young” at home.

…the need to further understand depression in relation to goal failure and coping cannot be better emphasized by the temporal overlap.

4. I appreciate the revision has added more quotations to reveal the cultural significance, for example: “High societal expectations of success”, and “Unable to pursue personal goals that were incongruent with familial and societal expectations”. However, would it possibly relate to the families and the society’s insufficient mental health literacy about depression? It’s suggested to refer existing evidence on the family illness perception and stigma as follows. Please add some related implication as well.

Huang CH, Li SM, & Shu BC. Exploring the relationship between illness perceptions and negative emotions in relatives of people with schizophrenia within the context of an affiliate stigma model. J. Nurs. Res 24(3), 217- 223 (2016).

Lee, S. K., Lin, E. C. L., Chang, Y. F., Shao, W. C., & Lu, R. B. (2017). Psychometric evaluation of family illness perceptions of patients with schizophrenia. Neuropsychiatry, 7(7), 739-747.

7. PLOS authors have the option to publish the peer review history of their article (what does this mean?). If published, this will include your full peer review and any attached files.

Reviewer #1: No

Reviewer #2: No

---

## [Author Response · Author response to Decision Letter 1]

10 Mar 2021

Dear Reviewers,

Thank you for taking the time to review our manuscript the second time. We have incorporated your comments and suggestions in our next draft. In addition, please find our point-by-point responses to your comments in the boxed text area below:

--

Reviewer #1: Thank you for responding to my comments. While the new version is substantially better, in my view there are still important aspects that are problematic, from an epistemological point of view. In addition, I still have some concerns on the compatibility of theory (based on a model stemming from a rather cognitive tradition) and methodology (based on understanding of contextualized experiences). Last, the discussion remains too narrow (scientific and practical implications are scarcely discussed).

General comment on epistemological concerns

The use of qualitative approaches in psychology does not imply the ‘mere’ implementation of interviews + conducting a thematic content analysis, but rather requires an in-depth reflection on how personal experiences are shaped by the social and the cultural, as well as more psychological and embodied aspects. In order to analyse individuals’ experiences, qualitative researches need to de-construct certain presuppositions (e.g. when you state ‘Since illness beliefs are personal in nature’ – this is a very strong assumption that many qualitative psychologists would argue against – as many approaches show how individual experiences and subjectivity – including ‘beliefs’ are socially constructed and therefore contextual – thus the opposite of ‘personal in nature’)

Reply: Perhaps a better way to explain is that individuals from collectivistic cultures have different communication styles than individualistic cultures. This has been rewritten in the next draft as we understand the possible conflict it brings theoretically. 

Another example of ‘lack of coherence’ between incompatible paradigms is the following statement: “From the perspectives of personality psychology and evolutionary biology, depressive symptoms are adaptive forms of coping that facilitates the detachment of existing unattainable goals to conserve resources which may otherwise further deplete such resources if individuals persisted with such goals (28-30).” This purpose is not coherent with qualitative approaches as it ‘essentialises’ goal setting as if this skill was completely detached from social contexts, social discourses, etc.

Reply: We agree with the reviewer that this argument is inconsistent with the social constructivist perspective. Therefore, we have removed parts of the discussion section that were incongruent.

General comment on the articulation of theory and qualitative methodology

As I have tried to explain, qualitative research tradition in psychology stresses the social embeddedness of psychological phenomena. Also, it requires to overcome the risk of methodolatry (see Chamberlain’s paper on this). There shall be more reasons than ‘simply’ practical or resource-related, to justify the use of a method rather than another one. Moreover, the difference that you state between interviews and focus groups is not necessarily true – many authors would argue that this statement

“Since illness beliefs are personal in nature one-to-one interviews are preferred over

focus group discussions as they provide the space to discuss personal attitudes without being pressured to give socially desirable answers” is not the difference between these two methods at all. As authors admit they come from more quantitative traditions, I would urge them to read qualitative research handbooks on these methods of data collection (and foremost, data analysis methods as they scarcely quote any authors in their analysis technique description)

Reply: In the previous and subsequent draft, we had cited Clarke and Braun, as well as the framework documented by Feredey and Muir-Cochrane as our primary texts. The article by Feredey and Muir-Cochrane describes the inductive coding technique by Boyatzis (1998). These are the two main texts that we have cited as we believe they are sufficient to inform and guide our thematic analyses.

General comment on scientific/broader implications

The discussion remains descriptive and too narrow – the reader may ask her/himself ‘so what?’ – what are the implications of studying individual experiences of depression in Singapour? As I had suggested in my previous review, the paper would improve its potential if it could broaden its scope from a theoretical perspective (What do your findings mean for theory on depression (Western-based) and furthermore, what about more practical implications for people who suffer from depression, for healthcare, for psychologists – namely in non-Western contexts like Singapour.

Reply: The discussion section has been revised extensively on this aspect as we agree it is still lacking in its current form. We believe the results will be useful in stressing the importance of the sociocultural context in the recovery of depressive disorders in young adults. We thank the reviewer for taking the time to review our manuscript. 

--

Reviewer #2: PONE-D-20-07514 A reduced state of being: illness perceptions in young adults diagnosed with depressive disorders

Thank you for asking me to secondly review this manuscript. Most of the comments has been well responded, but the following issues still need to be considered to make their effort valuable in cultural diversity and future implication.

1. I’m curious could it be possible that the illness perceptions and experience of your participants have not limited to the CSM framework? The author replied that their interview guide follows the CSM theory, that will easily produce an accordant result with the CSM as their prior framework. How could the researcher prevent yourselves to be mainly influenced by the prior understanding on the existing knowledge on the CSM?\\

Reply: The reviewer is accurate in stating that we are influenced by the literature reviews and CSM framework. While we recognized we are unable to distance ourselves entirely from being influenced by the theory, we had exercised a level of reflexivity that we believed were sufficient. For instance, the first author had written memos detailing her immediate thoughts during the entire process which were discussed alongside the development of the codebook among team members. Additionally, some members were not involved in the initial conceptualization of the study and hence were not informed of the literature reviews or framework. Thus, they had approached the interviews and analyses as ‘blinded’ co-investigators. Finally, interviewers conducted debriefings with their note-takers (if any) about the initial feelings and findings that had surfaced during the interview.

2. The stigma-related issues in non-western society have been mentioned in Discussion; but it has not been reflected from any of patients’ quotations?

Reply: While the cultural dimension plays an important role in illness perceptions of mental illness, the themes were too extensive to be reported in this manuscript. Cultural expectations that were directly related to the individual and illness perceptions, such as high societal expectations of success, were reported in this manuscript, whereas themes (and quotes/codes) stemming from the family/society that was more related to stigma than illness perception, will be reported in a separate manuscript. 

3. Please clarify some unclear wordings in Discussion, such as:

…that was heavily instilled “from young” at home.

…the need to further understand depression in relation to goal failure and coping cannot be better emphasized by the temporal overlap.

Reply: These portions have been edited to improve readability and we hope it reads better in the next draft. We thank the reviewer for highlighting these areas.

4. I appreciate the revision has added more quotations to reveal the cultural significance, for example: “High societal expectations of success”, and “Unable to pursue personal goals that were incongruent with familial and societal expectations”. 

Reply: Thank you for the suggestion, we had included more quotes for these themes and subthemes. 

However, would it possibly relate to the families and the society’s insufficient mental health literacy about depression? It’s suggested to refer existing evidence on the family illness perception and stigma as follows. Please add some related implication as well.

Huang CH, Li SM, & Shu BC. Exploring the relationship between illness perceptions and negative emotions in relatives of people with schizophrenia within the context of an affiliate stigma model. J. Nurs. Res 24(3), 217- 223 (2016).

Lee, S. K., Lin, E. C. L., Chang, Y. F., Shao, W. C., & Lu, R. B. (2017). Psychometric evaluation of family illness perceptions of patients with schizophrenia. Neuropsychiatry, 7(7), 739-747.

Reply: While we agree there is overlap between stigma and illness perception, we had decided to report the theme of stigma in a separate article since it is too extensive to be reported in a single manuscript. Most subthemes of stigma that we had found do not overlap with illness perceptions. For instance, a quote from participant C02 who is 20/Female/Chinese, “they (the family) believe in like those zodiac kind like maybe it’s your bad year. I don’t know. I really... I’m not like those, I’m not superstitious or like believe...I don’t believe in those Chinese, those tradition”, suggests that these religious attitudes that were maintained by the older generations were rejected by the participants. In this manuscript, we had reported only the specific cultural expectations that individuals with depression had articulated to have a direct contribution and relevance to the perception of their illness.

Both Huang et al. and Lee et al. examined stigma in relatives of patients with schizophrenia and found that stigma influenced their illness perception toward mental illness. These articles are important as they show how familial attitudes can implicate illness recovery. However, these articles report cross-sectional stigma from the perspective of family and thus may have few direct relevance. Next, mental health literacy and stigma in Taiwan and in Singapore are relatively similar (i.e.: Tonsing, K. N. (2018). A review of mental health literacy in Singapore. Social work in health care,57(1), 27-47.) and therefore we have noted these articles during the drafting of our other manuscript. We thank the reviewer for suggesting these articles as it is relevant to stigma, which we, as an organisation are also highly interested.

Thank you.

Warm regards,

Wen Lin

---

## [Decision Letter · Decision Letter 2]

3 May 2021

PONE-D-20-07514R2

A reduced state of being: illness perceptions in young adults diagnosed with depressive disorders

PLOS ONE

Dear Dr. Wen Lin Teh,

Thank you for submitting your manuscript to PLOS ONE. After careful consideration, we feel that it has merit but does not fully meet PLOS ONE’s publication criteria as it currently stands. Therefore, we invite you to submit a revised version of the manuscript that addresses the points raised during the review process.

As you will see, the reviewers give quite contradictory comments and recommendations. Unfortunately, we had to invite new reviewers for the revised version of your manuscript, this might explain some views different from thjose of the previous reviewers. I tend to follow reviewer 4 and not reviewer 3. However, I would like to ask you to implement as much of comments and suggestions of both reviewers.

We look forward to receiving your revised manuscript.

Kind regards,

Stephan Doering, M.D.

Academic Editor

PLOS ONE

Journal Requirements:

Reviewers' comments:

Reviewer's Responses to Questions

**Comments to the Author**

1. If the authors have adequately addressed your comments raised in a previous round of review and you feel that this manuscript is now acceptable for publication, you may indicate that here to bypass the “Comments to the Author” section, enter your conflict of interest statement in the “Confidential to Editor” section, and submit your "Accept" recommendation.

Reviewer #3: (No Response)

Reviewer #4: All comments have been addressed

2. Is the manuscript technically sound, and do the data support the conclusions?

Reviewer #3: Partly

Reviewer #4: Yes

3. Has the statistical analysis been performed appropriately and rigorously? 

Reviewer #3: N/A

Reviewer #4: Yes

4. Have the authors made all data underlying the findings in their manuscript fully available?

Reviewer #3: Yes

Reviewer #4: Yes

5. Is the manuscript presented in an intelligible fashion and written in standard English?

Reviewer #3: Yes

Reviewer #4: Yes

6. Review Comments to the Author

Reviewer #3: Thank you very much for the invitiation to review this interesting manuscript. I have noticed that both the previous review-ers as well as the authors have done a great job in improving the draft. However, I have tried to review the current version as an independent reviewer, and thus focussed my review on the manuscript rather than the previous comments.

The qualitative paper describes illness perceptions of Chinese, Malay and Indian young adults living in Singapure with depres-sive disorder. Using face-to-face semir-structured interviews, the authors concluded that participants experienced depres-sion as a reduced state of being. Further, five themes were extracted from thematic analysis. Both topic and study popula-tion seem of interest for the field, yet while I am not an expert in qualitative research and I did not review the first draft of the manuscript, I have some substantial issues with its current version - especially regarding its theoretical foundation and its overall scope.

Introduction

1.) In the introduction, the CSM is explained quite well. But the illness representations of mental illness are oversimpli-fied, and studies on bipolar disorder, schizophrenia and later anorexia nervosa combined to construct a narrative stating that “the boundary between mental illness and the self is far less clear [than in case of physical illnesses] and often intersects”. This argumentation completely neglects modern biopsychosocial approaches in medicine, foster-ing dualistic thinking. Second, egosyntonic and egodystonic concepts of different mental illnesses – including de-pression – have been discussed in clinical psychology for years.

2.) The overview of the existing qualitative literature on the topic also seems somewhat random. There are numerous qualitative studies on depression, for example, while the manuscript suggests that “the majority of qualitative work has been conducted amongst severe but less common forms of mental illness such as psychosis or schizophrenia”. This might be true with regard to illness representations, it is however important to be more precise with the re-search question you want to investigate in the manuscript, and what previous research you built your argumenta-tion upon.

3.) When I was reading the sentence “To the best of our knowledge, there has been no qualitative inquiry into illness perceptions of depression that is specific to a non-western psychiatric population residing in Singapore”, I felt like this is the main novelty the manuscript provides. However, reading the title as well as the abstract, I got a com-pletely different first impression of the manuscript’s aims and hypotheses. I think you should really focus on the main research question, rather than generalize qualitative results from a very specific population. For example, I found the description of the multi-ethnic Singapore population really interesting and as an important context factor for the interviews.

4.) I do not really understand from the introduction why you focused on young adults, apart from them being the “most pervasive” age group. I think a group of young people living in Singapor suffering from depression is a very in-teresting, yet also very specific group of patients. I think your work would benefit a lot from focusing on the particu-lar cultural and social insights you may get from this group, rather than trying to generalize the results on depres-sion (or even mental illness) as an entity.

5.) The first reseach question if far too general, and there are numerous (clinical) works on the illness perceptions in (young) persons with depression. As you are not able to answer it with your data, I would at least add “in Singapore” to it, and then elaborate on specific cultural influences in the second research question.

6.) I found the last paragraph in the introduction (your hypotheses?) to be very confusing, as you only give general statements rather than linking them to either your research questions or your data.

Methods

7.) Please state what “IMH” stand for.

8.) I am rather confued by the interviews taking place “at a convient place and time”, but then again in the research in-terview room in IMH “to minimize the possibility that participants may describe their illness beliefs in medical terms”. Also, you matched interviewers and patients by ethnicity, without providing any rational why. Further, in-clusion criteria included ability to read and speak English, yet interviews were only “primarily” conducted in English. While there might be many good and interesting reasons to conduct the qualitative interviews the way you did, from a scientific point of view it all sounds very arbitrary.

9.) I think you should provide some information on the interviewers as well (medical doctors, researchers, psycholo-gists?).

10.) Were there any other (methodological) reasons for using thematic analysis by Clarke and Braun rather than the au-thors being familiar with it?

Results

11.) I think this section would highly benefit from a figure or an additional numbering, though this is mainly a visual re-mark.

12.) At no point you mentioned the fact that only two of your participants were married. Did this emerge as a topic throughout the interviews? If not, this should be addressed oin the discussion?

Discussion

13.) In accordance to 12, you should elaborate further on the specifics of your study population.

14.) I found it interesting to read that depression is “a chronic condition with no means of full recovery”; was this a gen-eral statement or a point of view maintained by all interviewees?

15.) I enjoyed the explanations on the peculiarities of Singaporian culture. As you have a very special study population, I would be careful regarding over-generalizing your findings, though – especially regarding clinical implications and therapy. Accordingly, I would elaborate further on the weaknesses and limitations of the study .

Reviewer #4: PONE-D-20-07514R2. A reduced state of being: illness perceptions in young adults diagnosed with

depressive disorders

This is a very interesting paper handling illness perceptions in young adults in a relatively understudied populations, which makes that the paper contributes to the literature. The paper has been reviewed before and the reviewers provide valuable suggestions which have been answered appropriately by the authors. As a newly, more recent added reviewer I will not go into the points raised before. I really appreciated the wide range of quotes of the participants given in the result section to support the categories reported on.

Minor points:

The paper’s impact on cultural significance is not reflected in the title of the manuscript. I suggest to revise the title to make the paper easier findable for potential interested readers.

Abstract: …surviving with depressive disorders… sounds unfamiliar to me. Do you mean ..remitted .. here?

Ending the discussion with a paragraph on the limitations of the study is not elegant and makes it more difficult for the reader to find the take home message of the paper. Please end the discussion with a concluding paragraph reflecting on the research questions given in the introduction of the manuscript (so without including speculations that cannot be derived from the current results).

7. PLOS authors have the option to publish the peer review history of their article (what does this mean?). If published, this will include your full peer review and any attached files.

Reviewer #3: No

Reviewer #4: No

---

## [Author Response · Author response to Decision Letter 2]

16 May 2021

Reviewer #3: Thank you very much for the invitiation to review this interesting manuscript. I have noticed that both the previous review-ers as well as the authors have done a great job in improving the draft. However, I have tried to review the current version as an independent reviewer, and thus focussed my review on the manuscript rather than the previous comments.

The qualitative paper describes illness perceptions of Chinese, Malay and Indian young adults living in Singapure with depres-sive disorder. Using face-to-face semir-structured interviews, the authors concluded that participants experienced depres-sion as a reduced state of being. Further, five themes were extracted from thematic analysis. Both topic and study popula-tion seem of interest for the field, yet while I am not an expert in qualitative research and I did not review the first draft of the manuscript, I have some substantial issues with its current version - especially regarding its theoretical foundation and its overall scope.

Introduction

1.) In the introduction, the CSM is explained quite well. But the illness representations of mental illness are oversimpli-fied, and studies on bipolar disorder, schizophrenia and later anorexia nervosa combined to construct a narrative stating that “the boundary between mental illness and the self is far less clear [than in case of physical illnesses] and often intersects”. This argumentation completely neglects modern biopsychosocial approaches in medicine, foster-ing dualistic thinking. Second, egosyntonic and egodystonic concepts of different mental illnesses – including de-pression – have been discussed in clinical psychology for years.

Answer: Thank you for the comments. The biopsychosocial model explains complex individual-environment interactions in relation to illness. Research generally supports the biopsychosocial model as a dominant framework of today and its use have been widespread. Narrative or lay-person models are similar to the biopsychosocial model in that it informs the biopsychosocial aspects of illness. While the biopsychosocial model is commonly used to explain the causality of illness using a three pronged approach, the CSM is used to explain how illness is perceived by the patients themselves. In the narratives, young adults generally endorsed the biopsychosocial model of depression, however, they placed greater emphasis and salience on sociocultural elements. From a social constructivist perspective, sociocultural elements of depression may inform egosyntonic and egodystonic concepts that are relevant to a multi-cultural non-western setting, which can be highly applicable locally since existing models are predominantly western-based. Qualitative approaches thus allow for the voice of culture to be heard. We respectfully disagree with the reviewer indicating that the CSM neglects the biopsychosocial model completely. Instead, both models do overlap, and we believe that the current focus on narratives can inform, substantially, in areas that the biopsychosocial model overlooks, such as culture.

2.) The overview of the existing qualitative literature on the topic also seems somewhat random. There are numerous qualitative studies on depression, for example, while the manuscript suggests that “the majority of qualitative work has been conducted amongst severe but less common forms of mental illness such as psychosis or schizophrenia”. This might be true with regard to illness representations, it is however important to be more precise with the re-search question you want to investigate in the manuscript, and what previous research you built your argumenta-tion upon.

Answer: While it may be true that there are numerous qualitative studies that had explored depression in its broadest form, qualitative inquiry into illness narratives of depression in young adults in South-east Asia are few and far between. Most of existing research on illness perception (cited) have been documented in western communities, and the extent to which it is relatable to a multi-ethnic non-western setting may be questioned. While we did expect to find illness narratives to be highly associated with the self and self-concepts as reported by past research, our research questions (and of past research) remained broad rather than specific to capture cultural nuances. Prompts were made to improve the specificity of our interview schedules but they may depend very much on the direction and content of narratives.

3.) When I was reading the sentence “To the best of our knowledge, there has been no qualitative inquiry into illness perceptions of depression that is specific to a non-western psychiatric population residing in Singapore”, I felt like this is the main novelty the manuscript provides. However, reading the title as well as the abstract, I got a com-pletely different first impression of the manuscript’s aims and hypotheses. I think you should really focus on the main research question, rather than generalize qualitative results from a very specific population. For example, I found the description of the multi-ethnic Singapore population really interesting and as an important context factor for the interviews.

Answer: We agree with the reviewer that the existing title does not encapsulate the intention and results of our study. We have made further changes to the title to capture our work better.

4.) I do not really understand from the introduction why you focused on young adults, apart from them being the “most pervasive” age group. I think a group of young people living in Singapor suffering from depression is a very in-teresting, yet also very specific group of patients. I think your work would benefit a lot from focusing on the particu-lar cultural and social insights you may get from this group, rather than trying to generalize the results on depres-sion (or even mental illness) as an entity.

Answer: Young adulthood is arguably the period where an individual is most productive in life and from an economic standpoint, mental illness can pose a significant threat to productivity and disease burden. This is crucial for Singapore as human capital is considered an important asset. Thus, we have focused on young adults due to this overlap. We have since elaborated our study’s rationale in the next draft. 

We have added 3 new references to support the claims regarding the importance of culture in treatment relevance for young adulthood, citation [31]: McGorry PD, Goldstone SD, Parker AG, Rickwood DJ, Hickie IB. Cultures for mental health care of young people: an Australian blueprint for reform. The Lancet Psychiatry. 2014;1(7):559-68; 

Human capital in Singapore, citation [32] Cheng YE. Cultural Politics of Education and Human Capital Formation: Learning to Labor in Singapore. In: Abebe T, Waters J, Skelton T, editors. Laboring and Learning. Singapore: Springer Singapore; 2017. p. 265-84; and

Treatment gaps in Singapore pertaining to mental illness: citation [34] Subramaniam, M., Abdin, E., Vaingankar, J. A., Shafie, S., Chua, H. C., Tan, W. M., ... & Chong, S. A. (2019). Minding the treatment gap: results of the Singapore Mental Health Study. Social psychiatry and psychiatric epidemiology, 1-10.

5.) The first reseach question if far too general, and there are numerous (clinical) works on the illness perceptions in (young) persons with depression. As you are not able to answer it with your data, I would at least add “in Singapore” to it, and then elaborate on specific cultural influences in the second research question.

Answer: We have included the terms “Singapore”, and “locally” to be more specific, we thank the reviewer for the suggestions.

6.) I found the last paragraph in the introduction (your hypotheses?) to be very confusing, as you only give general statements rather than linking them to either your research questions or your data.

Answer: Thank you for spotting this, while we do expect to find similar themes that were reported in past literature, we refrained from the terms hypothesis testing as the nature of the study is exploratory.

Methods

7.) Please state what “IMH” stand for.

Answer: Thank you for spotting this. The IMH stands for the Institute of Mental Health, the only public state psychiatric hospital in Singapore.

8.) I am rather confued by the interviews taking place “at a convient place and time”, but then again in the research in-terview room in IMH “to minimize the possibility that participants may describe their illness beliefs in medical terms”. Also, you matched interviewers and patients by ethnicity, without providing any rational why. Further, in-clusion criteria included ability to read and speak English, yet interviews were only “primarily” conducted in English. While there might be many good and interesting reasons to conduct the qualitative interviews the way you did, from a scientific point of view it all sounds very arbitrary.

Answer: Thank you for noting the contradictions. By default, researchers conducted the interviews in the research rooms that were situated away from the clinic. However, if a participant is unwilling, we would interview the participant at a time and place that is convenient for the him/her. The interviews were conducted in English, but participants, as with all young adults in Singapore, are bilingual, therefore, code-switching to Mandarin, Malay, Tamil, or other languages may happen during the interview. This brings us to our final point on cultural sensitivity. Interviewees were matched with interviewers as much as possible to minimize potential barriers from the lack of cultural sensitivity that may present itself during the interview. We have made changes to the method section to clear this contradiction and we hope it reads better in the next draft.

9.) I think you should provide some information on the interviewers as well (medical doctors, researchers, psycholo-gists?).

Answer: The authors (WLT, ES, LC, RK, FD, SS) are all quantitative researchers formal training (degrees in psychology/sociology at least), and we have had since received professional training and are experienced in the use of thematic analysis. Thank you for this suggestion. We have state this in the next draft.

10.) Were there any other (methodological) reasons for using thematic analysis by Clarke and Braun rather than the au-thors being familiar with it?

Answer: Yes, we chose this method of analysis because of the advantages it has for researchers working in healthcare. Thematic analysis has several advantages that is favourable for us in overcoming methodological constraints, such as a lack of time, resources, and expertise that are required for many other qualitative approaches. We have included these reasons in the next draft.

Results

11.) I think this section would highly benefit from a figure or an additional numbering, though this is mainly a visual re-mark.

Answer: We have included additional numberings in this section and we hope it helps to read better in the next draft.

12.) At no point you mentioned the fact that only two of your participants were married. Did this emerge as a topic throughout the interviews? If not, this should be addressed oin the discussion?

Answer: Thank you for this comment. In Singapore, the median age at the time of first marriage is around 30 years old for men and 29 years old for women. As the average age of our participants is approximately 26 years old, it was expected that being married would not be a common attribute. Further, we did not find emergent or significant themes related specifically to being married in this study as only a few of whom were married. As only the prominent themes were discussed, we had not reported them.

Discussion

13.) In accordance to 12, you should elaborate further on the specifics of your study population.

Answer: The specifics of our study population are elaborated in the “Participants” subsection under “Methods”. 

14.) I found it interesting to read that depression is “a chronic condition with no means of full recovery”; was this a gen-eral statement or a point of view maintained by all interviewees?

It was both. It was a general statement and point of view maintained by the majority of our interviewees regardless of ethnicity. They had initially hoped depression could be resolved in an finite period of time, but they had soon felt it was not the case.

15.) I enjoyed the explanations on the peculiarities of Singaporian culture. As you have a very special study population, I would be careful regarding over-generalizing your findings, though – especially regarding clinical implications and therapy. Accordingly, I would elaborate further on the weaknesses and limitations of the study .

Answer: Generalization is not the aim of qualitative research. As reviewer #1 had pointed out, “the aim of qualitative research is not generalization but on the contrary, considering variablity, contextualization, narratives, singularity, experience - therefore it is not a limitation from my view”, which we had agreed with. 

Thank you for taking the time to review our manuscript.

--

Reviewer #4: PONE-D-20-07514R2. A reduced state of being: illness perceptions in young adults diagnosed with depressive disorders

This is a very interesting paper handling illness perceptions in young adults in a relatively understudied populations, which makes that the paper contributes to the literature. The paper has been reviewed before and the reviewers provide valuable suggestions which have been answered appropriately by the authors. As a newly, more recent added reviewer I will not go into the points raised before. I really appreciated the wide range of quotes of the participants given in the result section to support the categories reported on.

Minor points:

The paper’s impact on cultural significance is not reflected in the title of the manuscript. I suggest to revise the title to make the paper easier findable for potential interested readers.

Answer: Thank you for this suggestion. We have made slight changes to better encapsulate the significance of culture in this manuscript.

Abstract: …surviving with depressive disorders… sounds unfamiliar to me. Do you mean ..remitted .. here?

Answer: Yes, we do mean remitted and have changed that term.

Ending the discussion with a paragraph on the limitations of the study is not elegant and makes it more difficult for the reader to find the take home message of the paper. Please end the discussion with a concluding paragraph reflecting on the research questions given in the introduction of the manuscript (so without including speculations that cannot be derived from the current results).

Answer: We have added a paragraph to properly conclude and summarise the study’s aims and findings. Thank you for taking the time to review our manuscript.

--

Thank you.

Sincerely,

Wen Lin

---

## [Editor Report · Decision Letter 3]

26 May 2021

A reduced state of being: the role of culture in illness perceptions of young adults diagnosed with depressive disorders in Singapore

PONE-D-20-07514R3

Dear Dr. Teh,

We’re pleased to inform you that your manuscript has been judged scientifically suitable for publication and will be formally accepted for publication once it meets all outstanding technical requirements.

Kind regards,

Stephan Doering, M.D.

Academic Editor

PLOS ONE

---

## [Editor Report · Acceptance letter]

1 Jun 2021

PONE-D-20-07514R3 

A reduced state of being: The role of culture in illness perceptions of young adults diagnosed with depressive disorders in Singapore 

Dear Dr. Teh:

I'm pleased to inform you that your manuscript has been deemed suitable for publication in PLOS ONE. Congratulations! Your manuscript is now with our production department. 

Kind regards, 

on behalf of

Professor Stephan Doering 

Academic Editor

PLOS ONE